# The PERK arm of the unfolded protein response regulates satellite cell-mediated skeletal muscle regeneration

Guangyan Xiong[1], Sajedah M Hindi[1], Aman K Mann[2], Yann S Gallot[1], Kyle R Bohnert[1], Douglas R Cavener[3], Scott R Whittemore[1,4], Ashok Kumar[1]*

[1]Department of Anatomical Sciences and Neurobiology, University of Louisville School of Medicine, Louisville, United States; [2]duPont Manual High School, Louisville, United States; [3]Eberly College of Science, Pennsylvania State University, University Park, United States; [4]Department of Neurological Surgery, University of Louisville School of Medicine, Louisville, United States

**Abstract** Regeneration of skeletal muscle in adults is mediated by satellite stem cells. Accumulation of misfolded proteins triggers endoplasmic reticulum stress that leads to unfolded protein response (UPR). The UPR is relayed to the cell through the activation of PERK, IRE1/XBP1, and ATF6. Here, we demonstrate that levels of PERK and IRE1 are increased in satellite cells upon muscle injury. Inhibition of PERK, but not the IRE1 arm of the UPR in satellite cells inhibits myofiber regeneration in adult mice. PERK is essential for the survival and differentiation of activated satellite cells into the myogenic lineage. Deletion of PERK causes hyper-activation of p38 MAPK during myogenesis. Blocking p38 MAPK activity improves the survival and differentiation of PERK-deficient satellite cells in vitro and muscle formation in vivo. Collectively, our results suggest that the PERK arm of the UPR plays a pivotal role in the regulation of satellite cell homeostasis during regenerative myogenesis.

*For correspondence: ashok.kumar@louisville.edu

## Introduction

Skeletal muscle exhibits a remarkable capacity for regeneration following damage that is attributed to a population of muscle precursor cells, termed satellite cells (*Relaix and Zammit, 2012*). In undamaged skeletal muscle, these mononucleated cells reside between the basement membrane and the sarcolemma in a quiescent state (*Bentzinger et al., 2012*; *Yin et al., 2013*). Following injury to myofibers, satellite cells rapidly become activated, proliferate, and then fuse either to form new muscle fibers or to repair damaged parts of existing muscle fibers. Moreover, a fraction of the activated satellite cell population escapes differentiation and restores the pool of quiescent satellite cells in newly formed skeletal muscle (*Yin et al., 2013*). Quiescent satellite cells express high levels of paired box 7 (Pax7) protein (*Relaix and Zammit, 2012*). Following specification to the myogenic lineage, the levels of Pax7 are repressed and the levels of Myf5, MyoD, and myogenin are concomitantly increased, giving rise to proliferative myoblasts, which eventually fuse with injured myofibers to accomplish regeneration (*Kuang and Rudnicki, 2008*; *Relaix and Zammit, 2012*; *Yin et al., 2013*).

Accumulating evidence suggests that satellite cell homeostasis and function in adult skeletal muscle is regulated through the activation of multiple pathways including Notch, Wnt, NF-κB, and JAK-STAT signaling (*Dumont et al., 2015*). Recent studies also suggest that mitogen-activated protein kinase (MAPK) pathways play an important role in satellite cell homeostasis and function. Activation of ERK1/2 and JNK1/2 signaling enhances the self-renewal of satellite cells through downstream

activation of the c-JUN transcription factor, which augments the expression of Pax7 in satellite cells (*Abou-Khalil et al., 2009*; *Hindi and Kumar, 2016*; *Ogura et al., 2015*; *Shi et al., 2013*). In contrast, activation of p38 MAPK inhibits self-renewal and promotes differentiation of satellite cells into myoblasts (*Brien et al., 2013*; *Lluís et al., 2006*; *Wang et al., 2008*). While p38 MAPK promotes later stages of myogenic differentiation, its untimely activation reduces the pool of satellite cells and their regenerative potential in many conditions including aging (*Bernet et al., 2014*; *Cosgrove et al., 2014*).

Satellite cell survival and function can be influenced by both extrinsic and intrinsic stresses (*Brack and Muñoz-Cánoves, 2016*). The endoplasmic reticulum (ER) is an essential organelle of mammalian cells involved in many functions, such as protein folding and secretion and calcium homeostasis. Prolonged accumulation of misfolded/unfolded proteins causes stress in the ER, which initiates an evolutionarily conserved intracellular signaling mechanism known as the unfolded protein response (UPR). The UPR is comprised of three signaling branches, which are initiated by three ER receptors: RNA-dependent protein kinase-like ER eukaryotic translation initiation factor 2 alpha kinase (PERK), inositol-requiring enzyme 1 (IRE1), and activating transcription factor 6 (ATF6) (*Hetz, 2012*; *Wang and Kaufman, 2014*; *Wu and Kaufman, 2006*). Under basal conditions, these proteins are maintained in a relatively dormant state through binding to glucose-regulated protein 78 (GRP78), a chaperone protein in the ER lumen. When unfolded or misfolded proteins accumulate in the ER lumen, GRP78 binds preferentially to the misfolded proteins and is therefore released from PERK, IRE1, and ATF6, leading to their activation (14, 15). Activation of the UPR causes translational reprogramming, in which protein synthesis is globally repressed and accompanied by the preferential synthesis of a specific subset of mRNAs whose protein products are required for responding to ER stress (*Wang and Kaufman, 2014*). While the primary role of UPR is to restore ER function, chronic unmitigated ER stress can also lead to apoptotic cell death (*Flamment et al., 2012*; *Hetz, 2012*; *Wang and Kaufman, 2014*; *Wu and Kaufman, 2006*).

ER stress-induced UPR appears to play an important role in myogenesis, evidenced by findings that the ATF6 arm of the UPR is activated during skeletal muscle development and mediates apoptosis of a subpopulation of myoblasts that may be susceptible to cellular stresses (*Nakanishi et al., 2005*). Pan-inhibition of ER stress using pharmacological compounds blocks apoptosis and myoblast differentiation (*Nakanishi et al., 2005*), whereas inducers of ER stress selectively eliminate vulnerable myoblasts that allows surviving cells to differentiate more efficiently into myotubes in cell cultures (*Nakanishi et al., 2007*). A recent study has shown that the PERK/eukaryotic translation initiation factor 2α (eIF2α) arm of the UPR may be required for maintaining satellite cells in a quiescent state in adult skeletal muscle (*Zismanov et al., 2016*). However, the role and mechanisms of action of various arms of the UPR in satellite cell-mediated regenerative myogenesis remain completely unknown.

In the present study, using genetic mouse models, we have investigated the role of individual UPR pathways in satellite cell-mediated skeletal muscle regeneration. Our results demonstrate that the mRNA levels of *Eif2ak3* (encoding PERK) and *Ern1* (encoding IRE1), but not *Atf6* (encoding ATF6), are increased in satellite cells upon skeletal muscle injury in adult mice. We demonstrate that PERK, but not X-box-binding protein 1 (XBP1, the major target of IRE1 endonuclease activity which activates UPR), is required for satellite cell function during skeletal muscle repair. Our results also suggest that PERK is required for the survival of satellite cells during muscle regeneration and their differentiation in vitro. Furthermore, we found that the inactivation of PERK leads to hyper-activation of p38 MAPK. Inhibition of p38 MAPK using molecular and pharmacological approaches improves survival and differentiation in PERK-deficient myogenic cells both in vitro and in vivo.

## Results

### Ablation of PERK in satellite cells inhibits skeletal muscle regeneration in adult mice

We first investigated how the expression of various markers of ER stress are affected in satellite cells upon skeletal muscle injury. A combination of cell surface markers (CD45⁻, CD31⁻, Ter119⁻, Sca-1⁻, and α7-integrin⁺) can be used to isolate satellite cells from naïve and injured skeletal muscle of mice (*Hindi et al., 2012*). To understand how the expression of various markers of ER stress are regulated

in satellite cells upon muscle injury, we injected both tibialis anterior (TA) and gastrocnemius (GA) muscles of WT mice with 1.2% BaCl$_2$ solution, a widely used myotoxin for experimental muscle injury in mice, as previously described (*Hindi and Kumar, 2016*; *Ogura et al., 2015*). Control muscles were injected with saline only. After 5d, the TA and GA muscles were isolated and the single cell suspension made was subjected to fluorescence-activated cell sorting (FACS) for the isolation of quiescent and activated satellite cells from uninjured and injured muscle, respectively (*Hindi and Kumar, 2016*; *Hindi et al., 2012*). The isolated satellite cells were analyzed by qRT-PCR to detect the relative mRNA levels of various ER stress markers. The mRNA levels of *Eif2ak3* (encoding PERK protein) and *Ern1* (encoding IRE1), and *Atf4* were significantly increased, whereas the mRNA levels of *Atf3* and *Ppp1r15a* (encoding GADD34). were significantly reduced in satellite cells of injured muscle compared to that of uninjured muscle (*Figure 1A*). In contrast, there was no significant difference in the mRNA levels of *Atf6*, *Ddit3* (encoding CHOP), or *Hspa5* (encoding GRP78) in satellite cells of uninjured and injured skeletal muscle (*Figure 1A*). A recently published study has demonstrated phosphorylation of PERK (pPERK) in satellite cells of uninjured muscle (*Zismanov et al., 2016*). Using a FACS-based intracellular protein detection assay, we sought to investigate whether pPERK is also present in activated satellite cells of injured skeletal muscle of mice. Single cell suspensions prepared from 5d-injured TA muscle of WT mice were analyzed by FACS for the expression of α7-integrin and the phosphorylated form of PERK (pPERK). Results showed that pPERK protein was expressed in the α7-integrin[+] satellite cells (*Figure 1B*).

We next sought to investigate the role of PERK in satellite cells during regenerative myogenesis in vivo. We crossed floxed *Eif2ak3* (*Eif2ak3$^{fl/fl}$*) mice with *Pax7-CreER* mice (a tamoxifen-inducible satellite cell specific Cre line) (*Lepper et al., 2009*) to generate *Eif2ak3$^{fl/fl}$;Pax7-CreER* mice. Since *Pax7-CreER* mice are knock-in mice in which the expression of *Pax7* is regulated by endogenous Pax7 promoter (10), we used 9-week old *Eif2ak3fl/fl;Pax7-CreER* mice and treated them with tamoxifen or vehicle (corn oil) alone to generate satellite cell-specific PERK knockout (henceforth P7:PERK KO) and control (Ctrl) mice, respectively. The P7:PERK KO mice were fed a tamoxifen containing chow for the entire duration of the experiment. One week after the first injection of vehicle or tamoxifen, TA muscle of Ctrl and P7:PERK KO mice was injected with 100 μl of 1.2% BaCl$_2$ solution to induce necrotic muscle injury. Muscle regeneration was evaluated at day 5 and 14 post-BaCl$_2$ injection (*Figure 1C*). There was no difference in the overall body weight (*Figure 1D*) or wet weight of uninjured TA muscle (*Figure 1E*) between Ctrl and P7:PERK KO mice. However, the wet weight of injured TA muscle was significantly reduced in P7:PERK KO mice compared to injured TA muscle of Ctrl mice at day 5 after BaCl$_2$-mediated injury suggesting deficit in muscle regeneration (*Figure 1F*).

We next prepared transverse cryosections of the TA muscle and performed Hematoxylin and Eosin (H&E) staining. Results showed that the regeneration of TA muscle was considerably diminished in P7:PERK KO mice compared to Ctrl mice at 5d post injury (*Figure 1G*). There was an apparent decrease in the number and the size of centronucleated myofibers and increase in the cellular infiltrate in P7:PERK KO mice compared to Ctrl mice. Morphometric analysis of 5d-injured TA muscle sections showed significant decrease in the average cross-sectional area (CSA) and minimal Feret's diameter of regenerating (centronucleated) myofibers in P7:PERK KO mice compared with Ctrl mice (*Figure 1H and I*). Moreover, the percentage of myofibers containing two or more centrally located nuclei was significantly reduced in injured TA muscle of P7:PERK KO mice compared with Ctrl mice (*Figure 1J*). A deficit in muscle regeneration in P7:PERK KO mice was also clearly evident at 14d after muscle injury (*Figure 1G*). By preparing single cell suspension from 5d-injured TA muscles followed by FACS analysis using antibody against CD45 (a marker for leukocytes), we also investigated whether the deletion of PERK in satellite cells affects the inflammatory immune response. Results showed that there was no significant difference in the percentage of CD45[+] cells in 5d-injured TA muscle of Ctrl and P7:PERK KO mice (*Figure 1K*). Collectively, these results suggest that PERK is required for satellite cell-mediated regeneration of adult skeletal muscle.

## Genetic ablation of XBP1 in satellite cells does not affect skeletal muscle regeneration in adult mice

In response to ER stress, IRE1 becomes activated by autophosphorylation which, through its endonuclease activity, promotes splicing of a 26-base intron from X-box-binding protein 1 (*Xbp1*) mRNA (*Flamment et al., 2012*). Spliced *Xbp1* (*Xbp1s*) increases ER chaperones and other components to assist in the folding capacity of the ER (*Tirasophon et al., 1998*). Indeed, XBP1 mediates most of

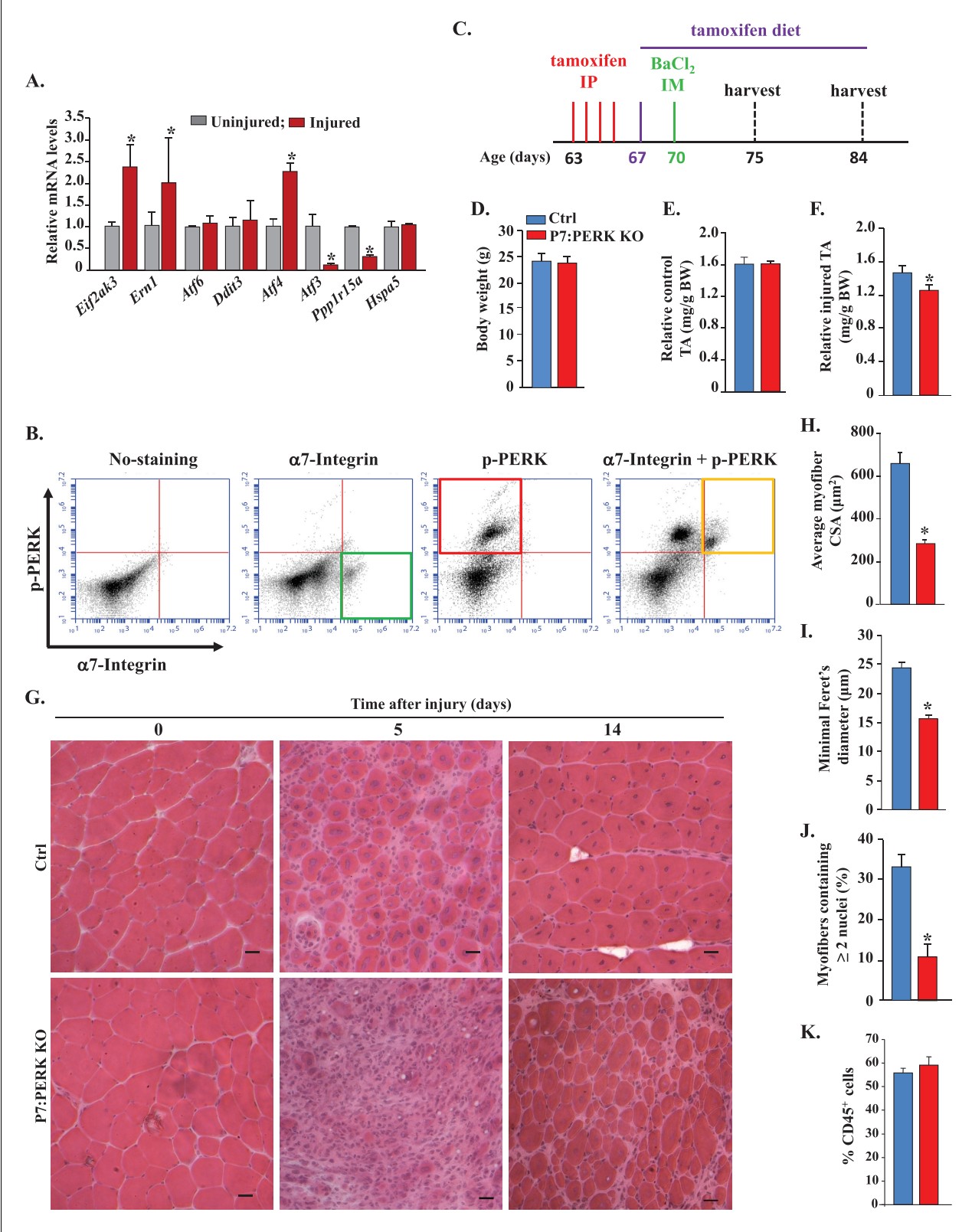

**Figure 1.** Role of PERK in satellite cell-mediated skeletal muscle regeneration. (**A**) Primary mononucleated cells were isolated from uninjured and 5d-injured hind limb muscle of WT mice. Satellite cells from cellular mixture were purified by FACS technique and immediately frozen. RNA was extracted and the transcript levels of the indicated ER stress markers quantified by qRT-PCR. N = 3 mice in each group. Data are mean ± SD. *p<0.05, values significantly different from uninjured muscle by unpaired t-test. (**B**) Primary mononucleated cells were isolated from the hind limb muscle of WT mice 5d

*Figure 1 continued on next page*

*Figure 1 continued*

after BaCl$_2$-mediated injury and subjected to FACS analysis for the expression of α7-integrin and phospho-PERK. Representative dot plots presented here demonstrate enrichment of phospho-PERK$^+$ cells amongst α7-integrin$^+$ population. N = 3 in each group. (C) Schematic representation of mice age and time of tamoxifen treatment and TA muscle injury and analysis. IP, intraperitoneal; IM, intramuscular. (D) Average overall body weight (BW) and (E) average uninjured TA muscle wet weight per gram BW of Ctrl and P7:PERK KO mice. TA muscle of Ctrl and P7:PERK KO mice were injured by intramuscular injection of 1.2% BaCl$_2$ solution. The muscles were harvested after 5d or 14d of muscle injury. (F) Average wet weight per gram BW of 5d-injured TA muscle of Ctrl and P7:PERK KO mice. (G) TA muscle sections were stained with H&E dye. Representative photomicrographs of H&E-stained sections illustrating a severe regeneration defect in injured TA muscle of P7:PERK KO mice compared with Ctrl littermates at day 5 (N = 6) and 14 (N = 3) after BaCl$_2$-mediated injury. Scale bar: 20 μm. Quantification of (H) average cross-sectional area (CSA) and (I) average minimal Feret's diameter of regenerating myofibers. (J) Percentage of myofibers containing two or more centrally located nuclei per field at day 5 post injury. (K) Percentage of CD45$^+$ cells in 5d-injured TA muscle of Ctrl (N = 4) and P7:PERK KO (N = 4) mice determined by FACS analysis. Data are mean ± SD. *p<0.05, values significantly different from corresponding Ctrl mice, as determined using unpaired Student's t-test.

the effects of IRE1 during ER stress (*Flamment et al., 2012*). To understand the role of IRE1/XBP1 arm of the UPR in satellite cell regenerative function, we crossed floxed *Xbp1* (*Xbp1$^{fl/fl}$*) mice with *Pax7-CreER* mice to generate *Xbp1$^{fl/fl}$;Pax7-CreER* mice. The *Xbp1$^{fl/fl}$;Pax7-CreER* mice were treated with tamoxifen or vehicle alone to generate satellite cell specific XBP1 knockout (henceforth P7: XBP1 KO) and control (Ctrl) mice, respectively. The TA muscle of these mice was subjected to BaCl$_2$-mediated injury. After 5d, muscle regeneration was monitored by performing H&E staining. Intriguingly, there was no significant difference in various parameters of muscle regeneration between Ctrl and P7:XBP1 KO mice (*Figure 2A–D*). These results suggest that PERK, but not the IRE1/XBP1 arm of the UPR, is required for regenerative function of satellite cells in adult skeletal muscle.

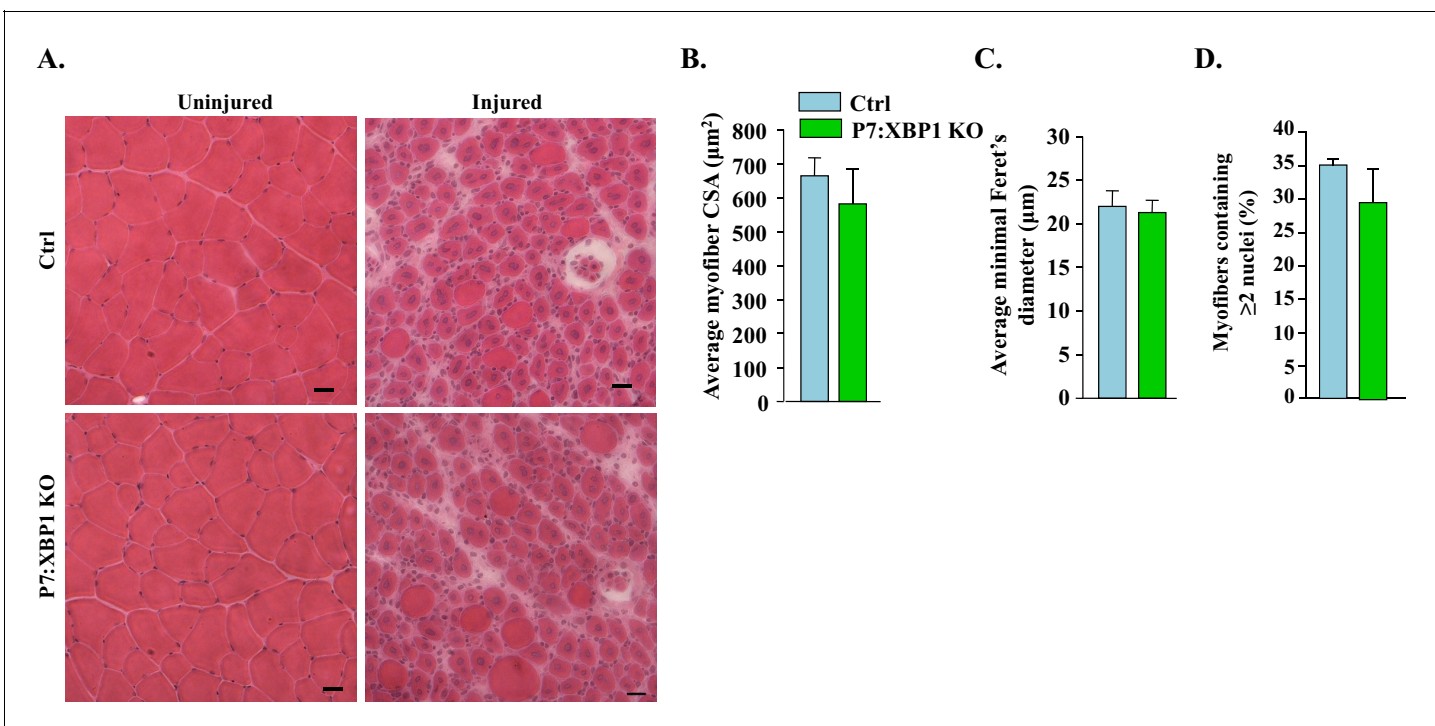

**Figure 2.** XBP1 is not required for satellite cell-mediated skeletal muscle regeneration. (A) TA muscle of Ctrl and P7:XBP1 KO mice were injured by intramuscular injection of 1.2% BaCl$_2$ solution. The muscles were harvested after 5d of muscle injury and sections were stained with H&E dye. Representative photomicrographs of H&E-stained sections suggesting no major effect on muscle regeneration between Ctrl and P7:XBP1 KO mice. Scale bar: 20 μm. Quantification of (B) average cross-sectional area (CSA), (C) average minimal Feret's diameter of regenerating myofibers, and (D) percentage of myofibers containing two or more centrally located nuclei per field at day 5 post injury. No significant differences were observed by unpaired Student's t-test between Ctrl and P7:XBP1 KO mice.

## Ablation of PERK in satellite cells diminishes the expression of early markers of muscle regeneration in mice

Skeletal muscle regeneration involves a hierarchical increase in the levels of myogenic regulatory factors Myf5, MyoD, and myogenin and the embryonic isoform of myosin heavy chain (eMyHC) (*Kuang and Rudnicki, 2008*; *Relaix and Zammit, 2012*; *Yin et al., 2013*). We investigated whether these markers of muscle regeneration are also affected in skeletal muscle of Ctrl and P7:PERK KO mice. We could not detect eMyHC$^+$ fibers in uninjured TA muscle of Ctrl or P7:PERK KO mice (data not shown). The appearance of eMyHC$^+$ myofibers was dramatically increased in TA muscle at 5d post-injury. However, the frequency of eMyHC$^+$ myofibers within laminin staining and the average minimal Feret's diameter of eMyHC$^+$ myofibers were significantly reduced in TA muscle of P7:PERK KO mice compared with corresponding Ctrl mice (*Figure 3A–C*). Moreover, qRT-PCR analysis confirmed that the mRNA levels of *Myh3* (transcript of eMyHC) were significantly reduced in injured TA muscle of P7:PERK KO mice compared with Ctrl mice (*Figure 3D*). Moreover, *Myf5*, *Myod1* (transcript of MyoD), and *Myog* (transcript of Myogenin) mRNA levels were significantly reduced in injured TA muscle of P7:PERK KO mice compared with injured TA muscle of Ctrl mice (*Figure 3E–G*). Immunoblot analysis also showed that the protein levels of eMyHC, MyoD, and myogenin were considerably reduced in injured TA muscle of P7:PERK KO mice compared with Ctrl mice (*Figure 3H*).

In a parallel experiment, we did not detect any significant difference in the *Myh3*, *Myod1*, and *Myog* mRNA levels between injured TA muscle of Ctrl and P7:XBP1 KO mice (*Figure 3I–K*). These findings suggest that PERK, but not IRE1/XBP1 arm of the UPR in satellite cells, is required for skeletal muscle regeneration in adult mice.

## Targeted deletion of PERK reduces the number of satellite cells in regenerating skeletal muscle

We next investigated whether deletion of PERK affects the abundance of satellite cells in skeletal muscle of mice. Transcription factor Pax7 is expressed in both quiescent and activated satellite cells and widely used as a marker to quantify satellite cells on skeletal muscle sections (*Relaix and Zammit, 2012*). TA muscle sections from Ctrl and P7:PERK KO were immunostained for Pax7 to detect satellite cells. The sections were also immunostained for laminin to mark the boundary of the myofibers. DAPI was used to identify nuclei. There was no significant difference in the abundance of satellite cells in uninjured TA muscle of Ctrl and P7:PERK KO mice (*Figure 4A and B*). A dramatic increase in the number of satellite cells was observed at day 5 post-BaCl$_2$-mediated injury. However, the frequency of satellite cells per myofiber was significantly reduced in injured TA muscle of P7: PERK KO mice compared with Ctrl mice (*Figure 4A and B*). Moreover, mRNA levels of *Pax7* were significantly reduced in injured TA muscle of P7:PERK KO mice compared to that of Ctrl mice (*Figure 4C*). Interestingly, there was a small but significant reduction in the mRNA levels of *Pax7* in uninjured TA muscle of P7:PERK KO compared to corresponding Ctrl mice (*Figure 4C*). By contrast, there was no significant difference in mRNA levels of *Pax7* in naïve condition or 5d-injured TA muscle of Ctrl and P7:XBP1 KO mice (data not shown) further suggesting that PERK but not the IRE1/XBP1 arm of the UPR is required for satellite cell homeostasis and expansion in skeletal muscle.

We next investigated how PERK regulates the fate of satellite cells. An ex vivo suspension culture of myofiber explants mimics muscle injury in vivo with respect to satellite cell activation, proliferation and initiation of myogenic differentiation (*Hindi and Kumar, 2016*; *Hindi et al., 2012*). In wild-type mice, immediately after isolation, each myofiber is associated with a fixed number of satellite cells which express Pax7, but not MyoD (Pax7$^+$/MyoD$^-$), that represent the quiescent satellite cell population. Upon culturing, satellite cells undergo multiple rounds of cell division through upregulating MyoD (Pax7$^+$/MyoD$^+$) and form cellular aggregates on myofibers. These cells then either self-renew (Pax7$^+$/MyoD$^-$) or commit to terminal differentiation (Pax7$^-$/MyoD$^+$) into the myogenic lineage (*Dumont et al., 2015*; *Hindi and Kumar, 2016*; *Hindi et al., 2012*). We prepared myofiber explants from extensor digitorum longus (EDL) muscles of Ctrl and P7:PERK KO mice and myofiber-associated satellite cells were analyzed immediately or after 72 hr of culturing by immunostaining with anti-Pax7 and anti-MyoD antibodies. There was no significant difference in the number of myofiber-associated Pax7$^+$/MyoD$^-$ cells immediately after isolation between Ctrl and P7:PERK KO mice. Moreover, none of the myofiber-associated cells stained positive for MyoD in either genotype at 0 hr

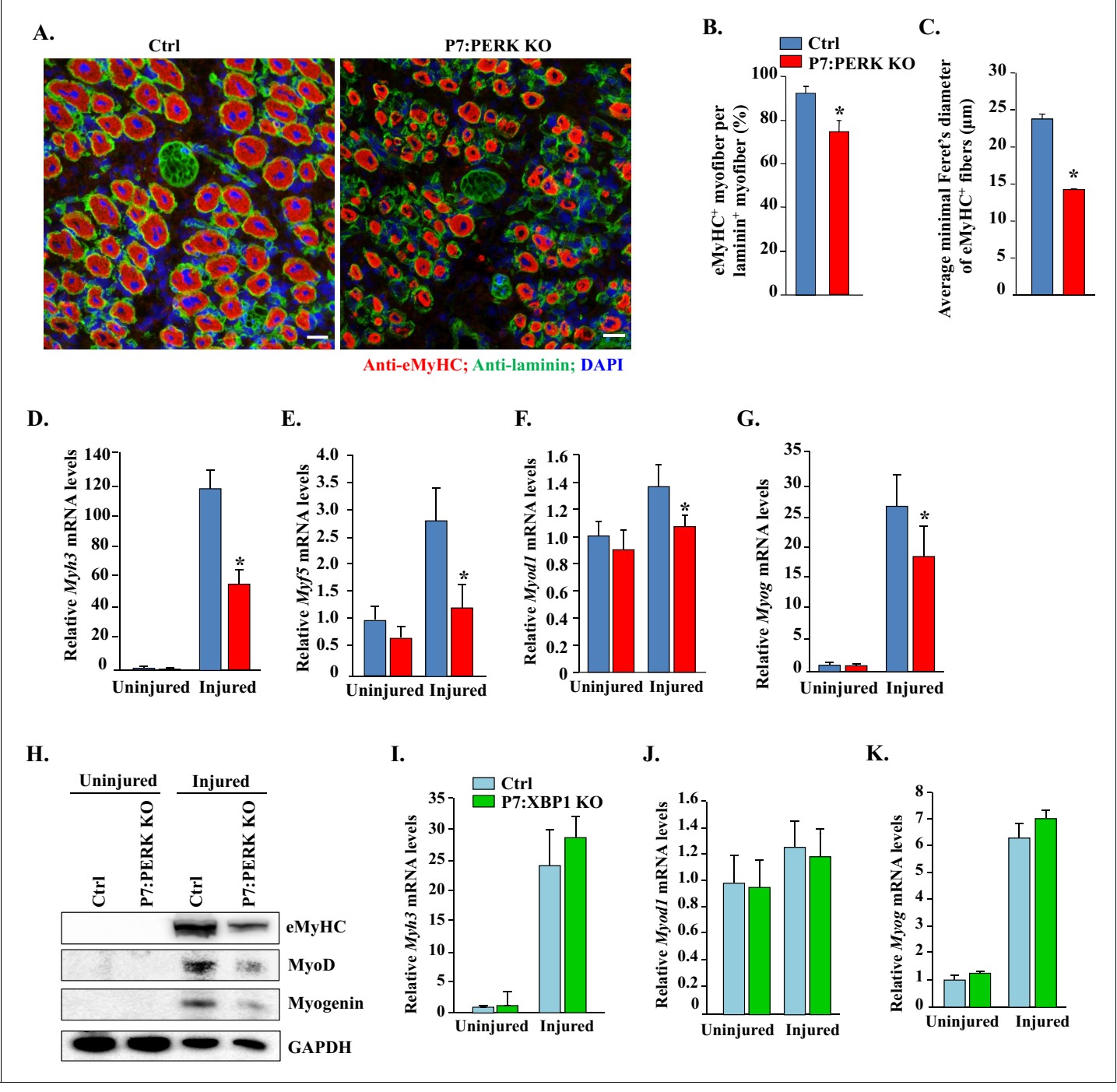

**Figure 3.** Deletion of PERK in satellite cells inhibits myofiber formation and expression of myogenic regulatory factors following injury. (A) Representative photomicrographs of 5d-injured TA muscle sections of Ctrl and P7:PERK KO mice after immunostaining for eMyHC (red color) and laminin (green). Nuclei were identified by staining with DAPI. Scale bar: 20 μm. (B) Percentage of eMyHC+ myofiber with laminin in 5d-injured TA muscle of Ctrl and P7:PERK KO mice. (C) Average minimal Feret's diameter of eMyHC+ myofibers of 5d-injured TA muscle of Ctrl and P7:PERK KO mice. Relative mRNA levels of (D) *Myh3*, (E) *Myf5*, (F) *Myod1*, and (G) *Myog* in uninjured and 5d-injured TA muscle of Ctrl and P7:PERK KO mice measured by performing qRT-PCR assay. N = 4 mice in each group for **A-G**. (H) Immunoblots presented here demonstrate the levels of eMyHC, MyoD, myogenin and an unrelated protein GAPDH in uninjured and injured TA muscle of Ctrl and P7:PERK KO mice. Relative mRNA levels of (I) *Myh3*, (J) *Myod1*, and (K) *Myog* in uninjured and 5d-injured TA muscle of Ctrl and P7:XBP1 KO mice measured by performing qRT-PCR assay. N = 4 mice in each group for each analysis. Data are mean ± SD. *p<0.05, values significantly different from corresponding injured TA muscle of Ctrl mice by unpaired t-test.

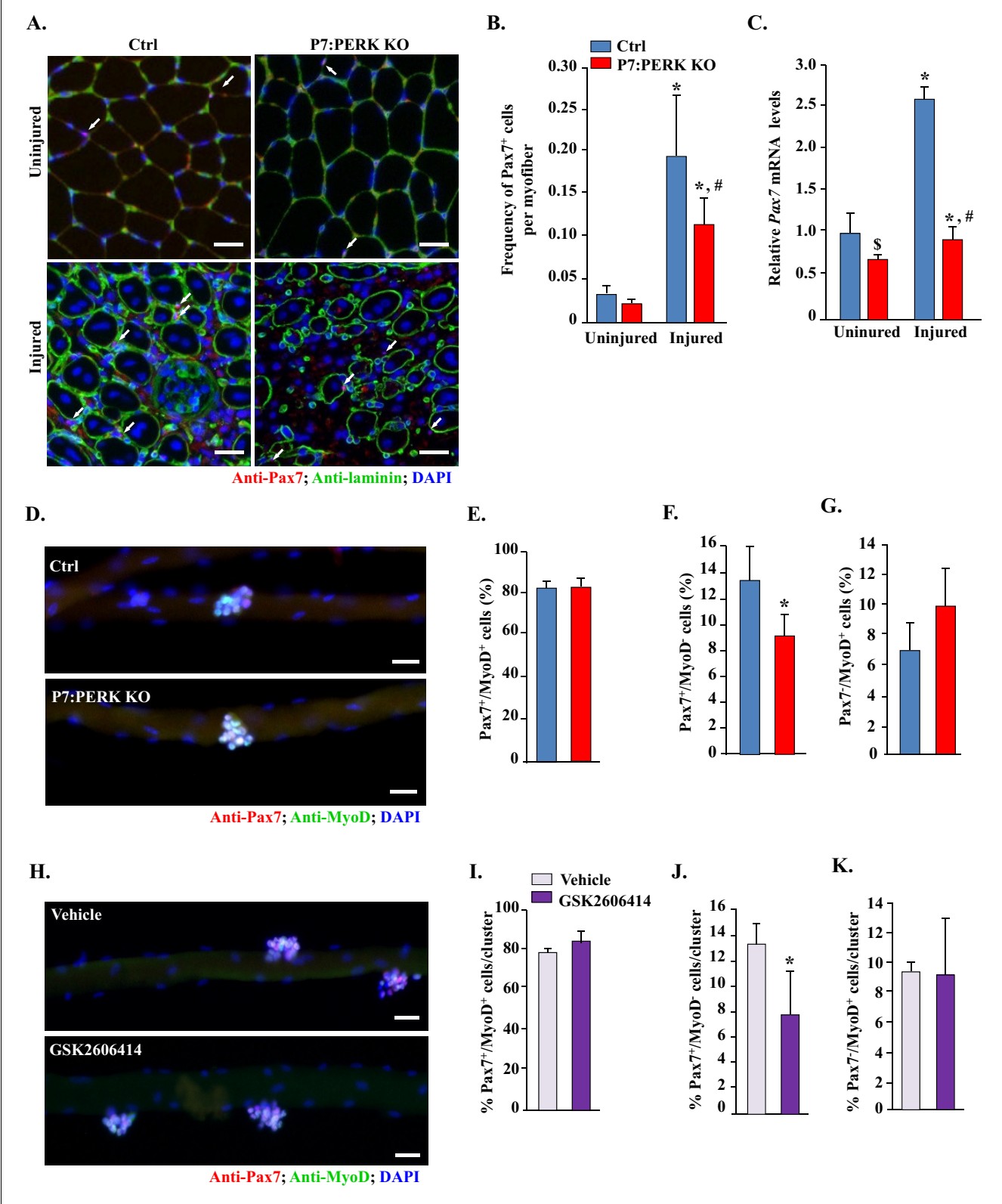

**Figure 4.** Inhibition of PERK reduces the number of Pax7+ cells during skeletal muscle regeneration and on cultured myofibers. (A) TA muscle of Ctrl and P7:PERK KO mice were injured by intramuscular injection of BaCl2. After 5d, uninjured and injured TA muscles were isolated and transverse sections made were analyzed by immunostaining for Pax7 and laminin. Nuclei were identified by co-staining with DAPI. Representative photomicrographs are shown here. Arrows point to Pax7+ cells. Scale bar: 50 μm. (B) Quantification of the frequency of Pax7+ cells per myofiber in

*Figure 4 continued on next page*

*Figure 4 continued*

uninjured and injured TA muscle section of Ctrl and P7:PERK KO mice. (C) Relative mRNA levels of *Pax7* in uninjured and 5d-injured TA muscle of Ctrl and P7:PERK KO mice measured by performing qRT-PCR. N = 4 in each group for **A–C**. *$p<0.05$, values significantly different from corresponding uninjured TA muscle of Ctrl or P7:PERK KO mice by unpaired *t*-test. #$p<0.05$, values significantly different from injured TA muscle of Ctrl mice by unpaired *t*-test. $$p<0.05$, values significantly different from uninjured TA muscle of Ctrl mice by unpaired *t*-test. (D) Single myofibers were isolated from EDL muscle of Ctrl and P7:PERK KO mice. After 72 hr of culturing, myofibers were collected and stained for Pax7 and MyoD. Representative merged images of cultured myofibers are presented here. Scale bars: 20 μm. Quantification of the percentage of (E) Pax7$^+$/MyoD$^+$, (F) Pax7$^-$/MyoD$^+$, and (G) Pax7$^+$/MyoD$^-$ cells per myofiber in Ctrl and P7:PERK KO cultures. (H) Single myofibers were isolated from EDL muscle of WT mice and treated with vehicle alone or 1 μM GSK2606414 for 72h. The myofibers were then collected and stained for Pax7 and MyoD. Nuclei were identified by staining with DAPI. Representative merged images of cultured myofibers incubated with vehicle alone or GSK2606414 for 72 hr are presented here. Scale bars: 20 μm. Quantification of the percentage of (I) Pax7$^+$/MyoD$^+$, (J) Pax7$^-$/MyoD$^+$, and (K) Pax7$^+$/MyoD$^-$ cells per myofiber in vehicle and GSK2606414-treated cultures. Analysis was done using 18–22 myofibers for each mouse. N = 3 mice in each group for **D-K**. Data are mean ± SD. *$p<0.05$, values significantly different from corresponding Ctrl or vehicle alone treated cultures by unpaired *t*-test.

The following figure supplements are available for figure 4:

**Figure supplement 1.** Deletion of PERK does not affect the number of Pax7$^+$ cells on freshly isolated myofibers.

**Figure supplement 2.** Inhibition of PERK reduces Pax7$^+$ cells in cultures.

**Figure supplement 3.** Targeted ablation of PERK reduces satellite cell count in injured muscle of mice.

(*Figure 4—figure supplement 1*). After 72 hr of culturing, myofiber-associated satellite cells formed clusters, a typical characteristic of activated satellite cells (*Figure 4D*). No significant difference was observed in the average number of clusters per myofiber or average number of cells per cluster between Ctrl and P7:PERK KO cultures (data not shown). There was no significant difference in Pax7$^+$/MyoD$^+$ cells on myofibers between Ctrl and P7:PERK KO cultures (*Figure 4E*). However, the percentage of Pax7$^+$/MyoD$^-$ cells was significantly reduced in P7:PERK KO cultures compared with Ctrl cultures (*Figure 4F*). There was also a trend towards increased proportion of Pax7$^-$/MyoD$^+$ cells in P7:PERK KO cultures compared with Ctrl cultures but it was not statistically significant (*Figure 4G*).

Using GSK2606414, a highly specific inhibitor of PERK phosphorylation (*Axten et al., 2012*, *2013*), we also investigated the effect of pharmacological inhibition of PERK on the fate of myofiber-associated satellite cells in cultures. For this experiment, suspension cultures of myofibers were established from the EDL muscle of WT mice followed by treatment with vehicle alone or GSK2606414 for 72 hr (*Figure 4H*). Consistent with the above results, GSK2606414 did not affect the average number of cellular clusters per myofiber or average number of cells per cluster on isolated single myofibers (data not shown). Inhibition of PERK using GSK2606414 had no effect on the proportion of Pax7$^+$/MyoD$^+$ cells (*Figure 4I*). By contrast, treatment with GSK2606414 significantly reduced the proportion of Pax7$^+$/MyoD$^-$ cells (*Figure 4J*) without having any significant effect on the proportion of Pax7$^-$/MyoD$^+$ cells (*Figure 4K*).

We also sought to investigate the effect of genetic ablation of PERK on self-renewal, proliferation, and differentiation of satellite cells in myofiber-free cultures. However, we could not purify enough number of satellite cells from hind limb muscle of P7:PERK KO mice for experimentation. While there was no apparent difference between initial number of satellite cells isolated from hind limb muscles of Ctrl and P7:PERK KO, most of P7:PERK KO satellite cells died within 3–4 days of culturing. Moreover, the surviving cells from P7:PERK KO mice failed to proliferate efficiently in culture. In one experiment, we plated surviving satellite cells from Ctrl and P7:PERK KO mice and monitored their proliferation by counting the number of cells per colony. There was a drastic reduction in the colony size in P7:PERK KO cultures compared to Ctrl cultures after 12d of their initial plating on the culture dish (*Figure 4—figure supplement 2*). It is notable that P7:PERK KO cells proliferate on isolated myofibers in cultures, but fail to survive or proliferate when cultured in myofiber-free conditions.

To investigate the effect of inhibition of PERK on satellite cells in myofiber-free cultures, we used a pharmacological approach. Primary myogenic cells prepared from WT mice were treated with vehicle alone or GSK2606414 for 24 hr followed by immunostaining for Pax7 and MyoD protein

(*Figure 4—figure supplement 2*). Most satellite cells in cultures were Pax7$^+$/MyoD$^+$, whereas a small proportion of them were Pax7$^+$/MyoD$^-$ or Pax7$^-$/MyoD$^+$. GSK2606414 had no significant effect on the proportion of Pax7$^+$/MyoD$^+$ or Pax7$^-$/MyoD$^+$. However, GSK2606414 significantly reduced the percentage of Pax7$^+$/MyoD$^-$ cells (*Figure 4—figure supplement 2*). Western blot analysis confirmed that GSK2606414 inhibited the phosphorylation without having any effect on the total levels of PERK protein in myogenic cultures (*Figure 4—figure supplement 2*).

Since we found a significant reduction in the proportion of Pax7$^+$/MyoD$^-$ satellite cells on cultured myofibers of P7:PERK KO mice compared to Ctrl mice, we next investigated whether deletion of PERK had a similar effect on satellite cells in injured muscle microenvironment. Muscle sections generated from 5d-injured TA muscle of Ctrl and P7:PERK KO mice were immunostained for Pax7 and MyoD protein whereas nuclei were identified by staining with DAPI. Consistent with published report (*Didier et al., 2012*), a vast majority of the nuclei including those within myofibers stained positive for MyoD which made it difficult to enumerate Pax7$^-$/MyoD$^+$ myogenic cells (*Figure 4—figure supplement 3A*). However, there were distinct populations of Pax7$^+$/MyoD$^-$ and Pax7$^+$/MyoD$^+$ cells within TA muscle sections. Our analysis showed that there was a significant reduction in the number of both Pax7$^+$/MyoD$^-$ and Pax7$^+$/MyoD$^+$ cells in TA muscle of P7:PERK KO mice compared to Ctrl mice (*Figure 4—figure supplement 3B and C*). It is notable that deletion of PERK did not affect the number of Pax7$^+$/MyoD$^+$ satellite cells in ex vivo myofiber cultures whereas a significant reduction in this cell population was noticeable in regenerating myofibers of P7:PERK KO mice. This could be attributed to the fact that myofibers in ex vivo cultures are maintained in growth conditions in which satellite cells do not undergo terminal differentiation and fusion. In contrast, in regenerating muscle, satellite cells are initially exposed to a pro-proliferation microenvironment followed by one that favors differentiation to conclude the regeneration program. These findings suggest that PERK is essential for maintaining the pool of satellite cells capable of undergoing self-renewal, proliferation, and fusion with injured myofibers.

## Ablation of PERK induces apoptosis in myogenic cells during regenerative myogenesis

Since the abundance of satellite cells was reduced in injured skeletal muscle of P7:PERK KO mice compared with Ctrl mice, we next investigated whether PERK regulates the survival of muscle progenitor cells during skeletal muscle regeneration. Transverse muscle sections generated from TA muscle of Ctrl and P7:PERK KO mice were stained with TUNEL to detect apoptotic cells within the basal lamina of myofibers. TUNEL$^+$ cells were not detected in uninjured TA muscle of Ctrl or P7: PERK KO mice (not shown). However, 5d-injured TA muscle contained several TUNEL$^+$ cells within the basal lamina in both Ctrl and P7:PERK KO mice (*Figure 5A*). Interestingly, the number of TUNEL$^+$ cells was significantly higher in injured TA muscle of P7:PERK KO mice compared with corresponding injured TA muscle of Ctrl mice (*Figure 5B*). We next used myofiber explants to study survival of satellite cells ex vivo. There were almost no TUNEL$^+$ cells on freshly isolated myofibers from EDL muscle of Ctrl or P7:PERK KO mice (data not shown). Surprisingly, the number of myofiber-associated TUNEL$^+$ cells was significantly increased in P7:PERK KO cultures compared to Ctrl cultures after 72 hr of culturing (*Figure 5C and D*).

## PERK is required for the survival of myogenic cells during in vitro differentiation

We first investigated how the levels of different components of PERK-mediated signaling are regulated during myogenic differentiation. Freshly prepared primary myogenic cells from WT mice were incubated in growth medium (GM) or differentiation medium (DM) for different time intervals. Consistent with published reports (*Nakanishi et al., 2005*; *Zismanov et al., 2016*), PERK and its downstream phosphorylation target eIF2α were highly phosphorylated in myogenic cells incubated in GM. Incubation of cells in DM led to reduced levels of phosphorylated PERK (p-PERK) and phosphorylated eIF2α (p-eIF2α) as well as total protein levels of their downstream target CHOP within 3 hr. However, the levels of p-PERK and p-eIF2α again started increasing by 6 hr, peaked at 12 hr, and remained elevated up to 24 hr after addition of DM. It is well established that differentiation-incompetent cells undergo apoptosis during in vitro myogenesis. Intriguingly, we found that phosphorylation of PERK and eIF2α (i.e. at 12 hr of addition of DM) was associated with concomitant increase in

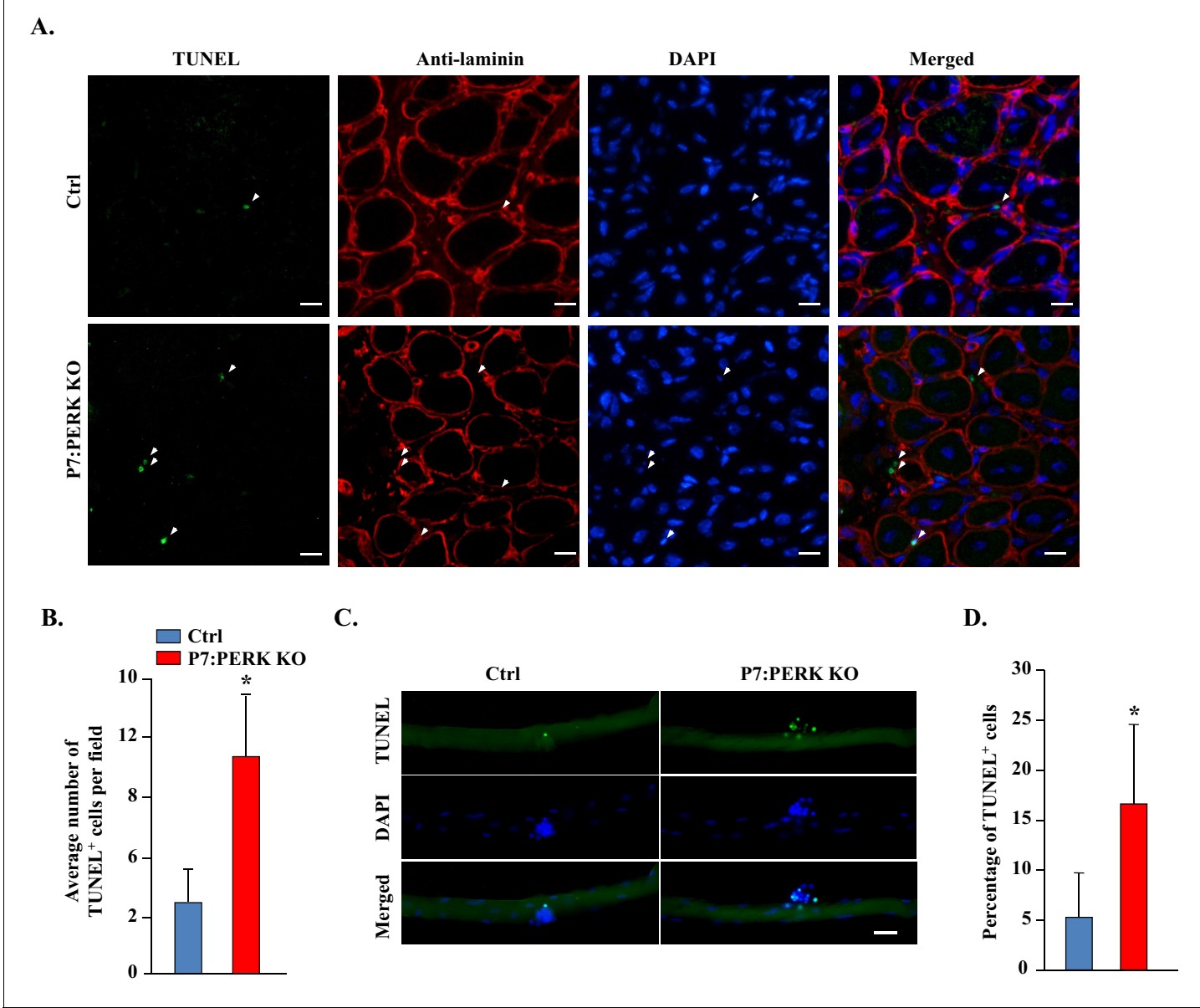

**Figure 5.** Deletion of PERK reduces survival of muscle progenitor cells during skeletal muscle regeneration and in myofiber explants. (**A**) Transverse sections prepared from 5d-injured TA muscle of Ctrl and P7:PERK KO mice were processed for the detection of TUNEL⁺ cells. The sections were also stained for laminin. Nuclei were counterstained with DAPI. Representative merged images are presented here. Arrows point to TUNEL⁺ cells within laminin staining around myofibers. Scale bar: 20 μm. (**B**) Quantification of number of TUNEL⁺ cells within laminin staining per field (~0.15 mm²) in 5d-injured TA muscle of Ctrl and P7:PERK KO mice. N = 4 mice in each group. (**C**) Single myofibers were isolated from EDL muscle of Ctrl and P7:PERK KO mice. After 72 hr of culturing, myofibers were collected and processed for TUNEL staining. Nuclei were counterstained with DAPI. Representative merged images of cultured myofibers are presented here. Scale bar: 20 μm. (**D**) Quantification of the percentage of TUNEL⁺ cells on myofibers in Ctrl and P7:PERK KO cultures. N = 3 mice in each group. Analysis was done using 14–18 myofibers for each mouse. Data are mean ± SD. *p<0.05, values significantly different from Ctrl mice or cultures by unpaired t-test.

the levels of cleaved caspase-3 and cleaved PARP proteins (*Figure 6A*), the markers of apoptosis which have been shown to be expressed and required in surviving differentiating myogenic cells upon incubation in DM (*Fernando et al., 2002*). While the pPERK levels were reduced after 24 hr, the levels of cleaved caspase-3 and cleaved PARP remained elevated even at 48 hr of addition of DM (*Figure 6A*). Although the physiological significance of these findings remains unknown, it is

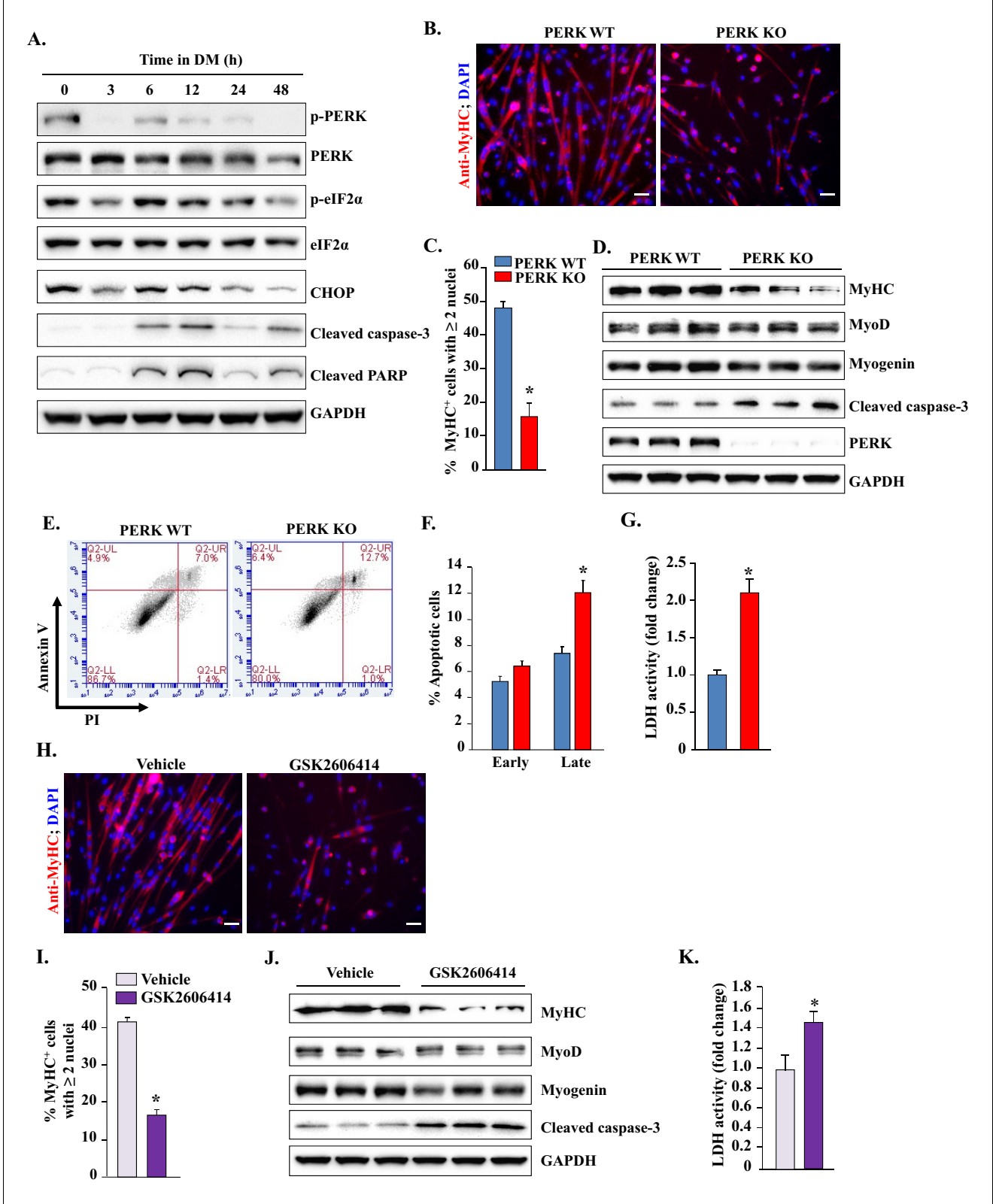

**Figure 6.** PERK is required for survival of myogenic cells during in vitro myogenesis. (**A**) Primary myogenic cells isolated from WT mice were incubated in DM for the indicated time intervals and protein extracts analyzed for the levels of various proteins related to the PERK arm of the UPR. Immunoblots presented here demonstrate the levels of phosphorylated and total PERK and eIF2α and total CHOP, cleaved PARP, cleaved caspase-3 and an unrelated protein GAPDH. (**B**) PERK WT and PERK KO myogenic cells were incubated in DM for 48 hr and the myotube formation was monitored by

*Figure 6 continued on next page*

*Figure 6 continued*

staining for MyHC. Nuclei were counterstained with DAPI. Representative images presented here demonstrate that myotube formation is considerably reduced in PERK KO cultures. Scale bar: 20 µm. (C) Quantification of percentage of MyHC⁺ myotubes containing two or more nuclei in PERK WT and PERK KO cultures after 48 hr of incubation in DM. (D) Immunoblots presented here show levels of MyHC, MyoD, myogenin, cleaved caspase-3, and unrelated protein GAPDH in PERK WT and PERK KO cultures after 48 hr of incubation in DM. (E) PERK WT and PERK KO myogenic cells were incubated in DM for 36–48 hr. Both adherent and floating cells were collected and stained for Annexin V and propidium iodide (PI) and analyzed by FACS to detect early and late apoptotic cells. Representative dot plots are presented here. (F) Quantification of early and late apoptotic cells in PERK WT and PERK KO cultures after FACS analysis. (G) Relative amounts of lactate dehydrogenase (LDH) in supernatants of PERK WT and PERK KO cultures after 36–48 hr of incubation in DM. (H) Primary myoblasts prepared from WT mice were treated with vehicle alone or 1 µM GSK2606414 for 48 hr and the myotube formation was monitored by immunostaining for MyHC. Nuclei were identified by staining with DAPI. Representative merged images are presented here. Scale bar: 20 µm. (I) Quantification of percentage of MyHC⁺ myotubes containing two or more nuclei in vehicle or GSK2606414-treated cultures after 48 hr of incubation in DM. (J) Protein levels of MyHC, MyoD, myogenin, cleaved caspase-3, and GAPDH in vehicle and GSK2606414-treated WT cultures 48 hr after incubation in DM. (K) Relative amounts of LDH in supernatants of vehicle alone and GSK2606414-treated cultures. N = 4 in each group. Data are mean ± SD. *p<0.05, values significantly different from corresponding PERK WT or vehicle alone cultures by unpaired t-test.

possible that the PERK arm of the UPR regulates the initiation of the differentiation program in myogenic cells after incubation in DM.

We next sought to determine whether PERK has any role in myogenic differentiation. Since satellite cells from P7:PERK KO mice did not grow in culture, we isolated satellite cells from Ctrl mice. After establishing the cultures, the cells were treated with vehicle alone or 0.5 µM tamoxifen (TAM) for 48 hr to generate PERK WT and PERK KO cells, respectively (*Hindi and Kumar, 2016*; *Ogura et al., 2015*). The cells were washed and incubated in GM for an additional 48 hr. In our initial experiments, we confirmed that 0.5 µM TAM has absolutely no effect on cellular viability or differentiation of myoblasts into multinucleated myotubes in WT cultures. Moreover, there was also no significant difference in cellular viability between PERK WT and PERK KO cells in GM (data not shown). Next, PERK WT and PERK KO cells were incubated in DM for 48 hr and the myotube formation was monitored by immunostaining for myosin heavy chain (MyHC) protein. DAPI was used to identify nuclei. Interestingly, myotube formation was considerably reduced in PERK KO cultures compared to PERK WT upon incubation in DM (*Figure 6B and C*). Western blot analysis showed that the levels of MyHC, but not MyoD and myogenin, were significantly reduced in PERK KO compared with PERK WT cultures (*Figure 6D*). Our analysis also showed that there was >90% reduction in PERK protein levels in PERK KO cultures compared with PERK WT cultures (*Figure 6D*). Intriguingly, we found that there was increased cell death in PERK KO compared to PERK WT cultures after incubation in DM, which was also confirmed by performing Annexin V and propidium iodide (PI) staining followed by FACS analysis (*Figure 6E and F*). Consistent with increased cell mortality, levels of cleaved caspase-3 were also found to be increased in PERK KO cultures compared to PERK WT cultures (*Figure 6D*). Lactate dehydrogenase (LDH) is a stable enzyme which is released in culture supernatants upon cell death. Results showed that the levels of LDH were significantly higher in culture supernatants of PERK KO cells compared to PERK WT at 48 hr after incubation in DM (*Figure 6G*), further suggesting that deletion of PERK reduces the survival of progenitor cells during myogenic differentiation.

To further evaluate the role of PERK in myogenesis, we next investigated the effect of pharmacological inhibition of PERK on the survival and differentiation of myogenic cells. Primary myogenic cultures prepared from hind limb muscle of WT mice were incubated in DM for 48 hr with vehicle alone or GSK2606414. Similar to PERK KO cultures, we observed reduced myotube formation in GSK2606414-treated cultures compared to cultures treated with vehicle alone (*Figure 6H and I*). Moreover, the levels of MyHC were considerably reduced, whereas the levels of cleaved caspase-3 were increased upon treatment of myogenic cells with GSK2606414 (*Figure 6J*). Furthermore, the levels of LDH were significantly increased in culture supernatants of GSK2606414-treated cells compared to those treated with vehicle alone (*Figure 6K*).

## Inhibition of PERK causes spurious activation of p38 MAPK in cultured myogenic cells

The activation of PERK in response to ER stress leads to the phosphorylation of eIF2α, which results in global inhibition of protein synthesis (*Hetz, 2012*). We investigated whether inhibition of PERK

using GSK2606414 affects the rate of protein synthesis in myogenic cells. Results showed that the inhibition of PERK considerably increased the rate of protein synthesis in myogenic cells in GM or 12 hr of incubation in DM (*Figure 7A*). Since myogenesis is regulated through the coordinated activation of multiple signaling pathways (*Dumont et al., 2015*), we next investigated whether the inhibition of PERK perturbs the activation of various signaling proteins in myogenic cells after incubation in DM. There was no difference in the levels of phosphorylated ERK1/2, Akt, or p65 (a marker of activation of canonical NF-κB pathway) protein between control and GSK2606414 treated myogenic cultures after 12 hr of incubation in DM. There was also no change in the relative levels of p100 and p52 proteins suggesting that the non-canonical NF-κB pathway was not affected upon inhibition of PERK in myogenic cells (*Figure 7B*). Interestingly, we found that levels of phosphorylated p38 (p-p38) MAPK were significantly increased in GSK2606414-treated cultures compared to those treated with vehicle alone (*Figure 7B and C*). Moreover, we found increased levels of p-p38 MAPK in PERK KO cultures compared with PERK WT cultures upon incubation in DM (*Figure 7D*). Using another approach, we investigated the effects of knockdown of PERK on the activation of p38 MAPK. We generated adenoviral vectors expressing scrambled (control) shRNA or PERK shRNA. As expected, we found knockdown of PERK reduced the levels of MyHC upon incubation in DM (*Figure 7E*). Importantly, we found that the levels of p-p38 MAPK were considerably increased upon knockdown of PERK in myogenic cells (*Figure 7E*). It is notable that the increased activation of p38MAPK was observed when cells were incubated in DM. There was no difference in the levels of p-p38 between control and GSK2606414-treated cultures in GM (data not shown). Taken together, these results suggest that the inhibition of PERK leads to hyper-activation of p38 MAPK during myogenic differentiation.

## Inhibition of p38 MAPK reduces mortality and improves myotube formation in PERK KO cultures

Activation of p38 MAPK can cause cell death and precocious differentiation of satellite cells (*Bernet et al., 2014*; *Brien et al., 2013*; *Cai et al., 2006*; *Cosgrove et al., 2014*). We investigated whether the increased activation of p38 MAPK is responsible for increased cell death and reduced myotube formation upon inhibition of PERK. MKK3 and MKK6 are the upstream kinases which directly phosphorylate p38 MAPK in mammalian cells (*Brancho et al., 2003*). To inhibit p38 MAPK, we transfected PERK WT and PERK KO myogenic cells with dominant negative (DN) mutants of MKK3 or MKK6. After 36 hr of transfection, the cells were incubated in DM for 48 hr and the myotube formation was measured. Interestingly, overexpression of DN-MKK3 or DN-MKK6 significantly improved myotube formation in PERK KO cultures (*Figure 8A and B*). Moreover, overexpression of DN-MKK3 or DN-MKK6 improved cell survival in PERK KO cultures during myogenic differentiation evidenced by the significant reduction in the levels of LDH in culture supernatants (*Figure 8C*). We also investigated the effect of pharmacological inhibition of p38 MAPK on cell survival and myotube formation in PERK KO cultures. Results showed that treatment of cells with SB202190, an inhibitor of p38 MAPK, dramatically improved the survival of cells in PERK KO cultures (*Figure 8D and E*). While we observed increased survival of MyHC$^+$ myotubes upon treatment with SB202190, the diameter of multinucleated myotubes was smaller in PERK KO cultures. Indeed, SB202190 also reduced size of myotubes in PERK WT cultures (*Figure 8D*). This is because p38 MAPK is also essential for terminal differentiation of myoblasts and SB202190 acutely inhibits the activity of various isoforms of p38 MAPK in mammalian cells. Collectively, these results suggest that PERK inhibition perturbs the activation of p38 MAPK which causes cell death and reduces the formation of multinucleated myotubes upon incubation in DM.

## Inhibition of p38 MAPK improves skeletal muscle regeneration in P7: PERK KO mice

Our preceding results showed that p38 is activated and responsible for cell death in PERK KO myogenic cultures during myogenic differentiation. We next investigated whether p38 MAPK is also activated in satellite cell of P7:PERK KO mice in response to muscle injury. TA muscle of Ctrl and P7: PERK KO mice was injected with 100 μl of 1.2% BaCl$_2$ and after 5d, the level of p-p38 MAPK in satellite cells was detected by FACS method similar to as described (*Ogura et al., 2015*). Our analysis showed that the amount of phosphorylated p38 MAPK was significantly higher in satellite cells of

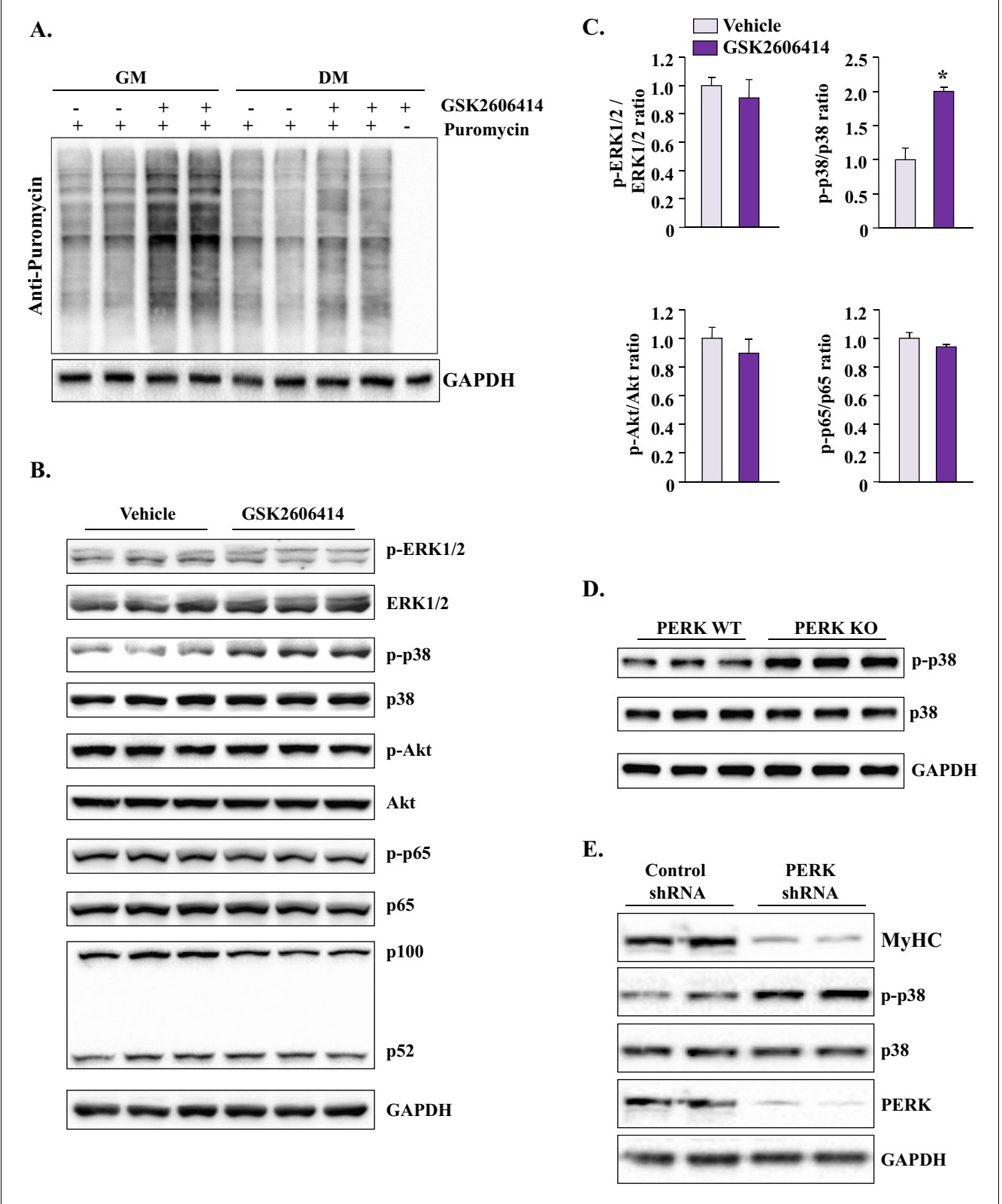

**Figure 7.** Inhibition of PERK induces protein synthesis and activates p38 MAPK in cultured myogenic cells. (**A**) Primary myogenic cells prepared from hind limb muscles of WT mice were incubated with growth medium (GM) or DM with or without 1 μM GSK2606414 for 12 hr. The rate of protein synthesis was measured by performing SUnSET assay. Representative immunoblot presented here demonstrates that inhibition of PERK increases the rate of protein synthesis in myogenic cultures. (**B**) Primary myogenic cells were incubated in DM for 12 hr with or without 1 μM GSK2606414 and

*Figure 7 continued on next page*

*Figure 7 continued*

processed by performing Western blot. Immunoblots presented here demonstrate phosphorylated and total levels of ERK1/2, p38MAPK, Akt, and p65 proteins and relative amounts of p100 and p52 proteins and GAPDH. (C) Ratio of phosphorylated vs total ERK1/2, p38 MAPK, Akt, and p65 in vehicle or GSK2606414-treated cultures measured by densitometric analysis of immunoblots. (D) Immunoblots presented here demonstrate the levels of phosphorylated and total p38 and total GAPDH protein in PERK WT and PERK KO cultures at 24 hr after incubation in DM. (E) Primary myogenic cultures were transduced with adenoviral vectors expressing a scrambled (control) shRNA or PERK shRNA. The cells were incubated in DM and the levels of MyHC, phosphorylated and total p38 MAPK, and total PERK, and GAPDH were measured by performing western blot. N = 3 in each group. Data are mean ± SD. *p<0.01, values significantly different from corresponding cultures treated with vehicle alone, as determined by a Student's unpaired t-test.

P7:PERK KO mice compared to Ctrl mice at day 5 post-muscle injury (*Figure 9A and B*). We also compared levels of phosphorylated p38 MAPK in injured TA muscle of Ctrl and P7:PERK KO mice. Results showed that the levels of p-p38 MAPK were significantly higher in regenerating TA muscle of P7:PERK KO mice compared with Ctrl mice (*Figure 9C and D*). We next investigated the effect of inhibition of p38 MAPK on muscle regeneration. TA muscle of Ctrl and P7:PERK KO mice was injured using 1.2% BaCl$_2$. The mice were also treated daily with vehicle alone or p38 MAPK inhibitor, SB202190. After 5 days, the TA muscle was isolated and analyzed by performing H&E staining. Interestingly, muscle regeneration was significantly improved in TA muscle of P7:PERK KO mice after treatment with SB202190 (*Figure 9E*). Morphometric analysis also showed that the average CSA and minimal Feret's diameter (*Figure 9F and G*) and number of myofibers containing two or more centronucleated myofibers were significantly increased in TA muscle of SB202190-treated P7:PERK KO mice compared to P7:PERK KO treated with vehicle alone (*Figure 9H*). These results suggest that increased activation of p38 MAPK is a potential mechanism for the inhibition of skeletal muscle regeneration upon injury in P7:PERK KO mice.

## Discussion

Many types of cellular stress cause accumulation of misfolded protein in the ER and consequent activation of the UPR enables the cell to either resolve the stress or initiate apoptosis. The UPR is comprised of the PERK, IRE1, and ATF6 pathways, which increase the protein folding capacity of the ER through augmenting the levels of chaperone proteins and attenuating global protein synthesis (*Hetz, 2012*; *Wang and Kaufman, 2014*; *Wu and Kaufman, 2006*). PERK is also involved in the termination of the UPR after the stress has been relieved through activation of GADD34, which dephosphorylates p-eIF2α (*Ron and Walter, 2007*). Recent studies have shown that the UPR has an important role in survival, self-renewal, proliferation, and differentiation of stem cells. For example, activation of PERK predisposes human hematopoietic stem cells (HSCs) to apoptosis, whereas closely related progenitors of HSCs exhibit an adaptive response leading to their survival (*van Galen et al., 2014*). In Drosophila, PERK is required for intestinal stem cell proliferation under both homeostatic and stress conditions (*Wang et al., 2015*). Stem cells are also one of the important cell types most susceptible to causing tumorigenesis in mammals. Forced activation of UPR induces the differentiation of colon cancer stem cells and makes them susceptible to conventional chemotherapy (*Wielenga et al., 2015*).

Skeletal muscle is a highly regenerative tissue, attributed to the presence of satellite stem cells. The regenerative capacity of skeletal muscle is often impaired in many chronic disease states, genetic muscle disorders, and during aging (*Dumont et al., 2015*). However, the role of individual arms of the UPR in the regulation of satellite cell function has not been fully elucidated. The induction of PERK and IRE1 in satellite cells upon skeletal muscle injury (*Figure 1A*) and our results with P7:PERK KO mice provides the first genetic evidence that the PERK arm of the UPR is critical for satellite cell survival and function during skeletal muscle regeneration. Satellite cell-specific deletion of PERK considerably reduced the early markers of skeletal muscle regeneration (*Figure 1* and *Figure 3*). Moreover, deletion of PERK reduced the number of satellite cells in response to muscle injury suggesting that the PERK arm of the UPR is essential for the survival or expansion of satellite cells during muscle regeneration in vivo (*Figure 4*). IRE1 is an endonuclease which mediates its most effects through alternative splicing of the XBP1 transcription factor during ER stress (*Yoshida et al.,*

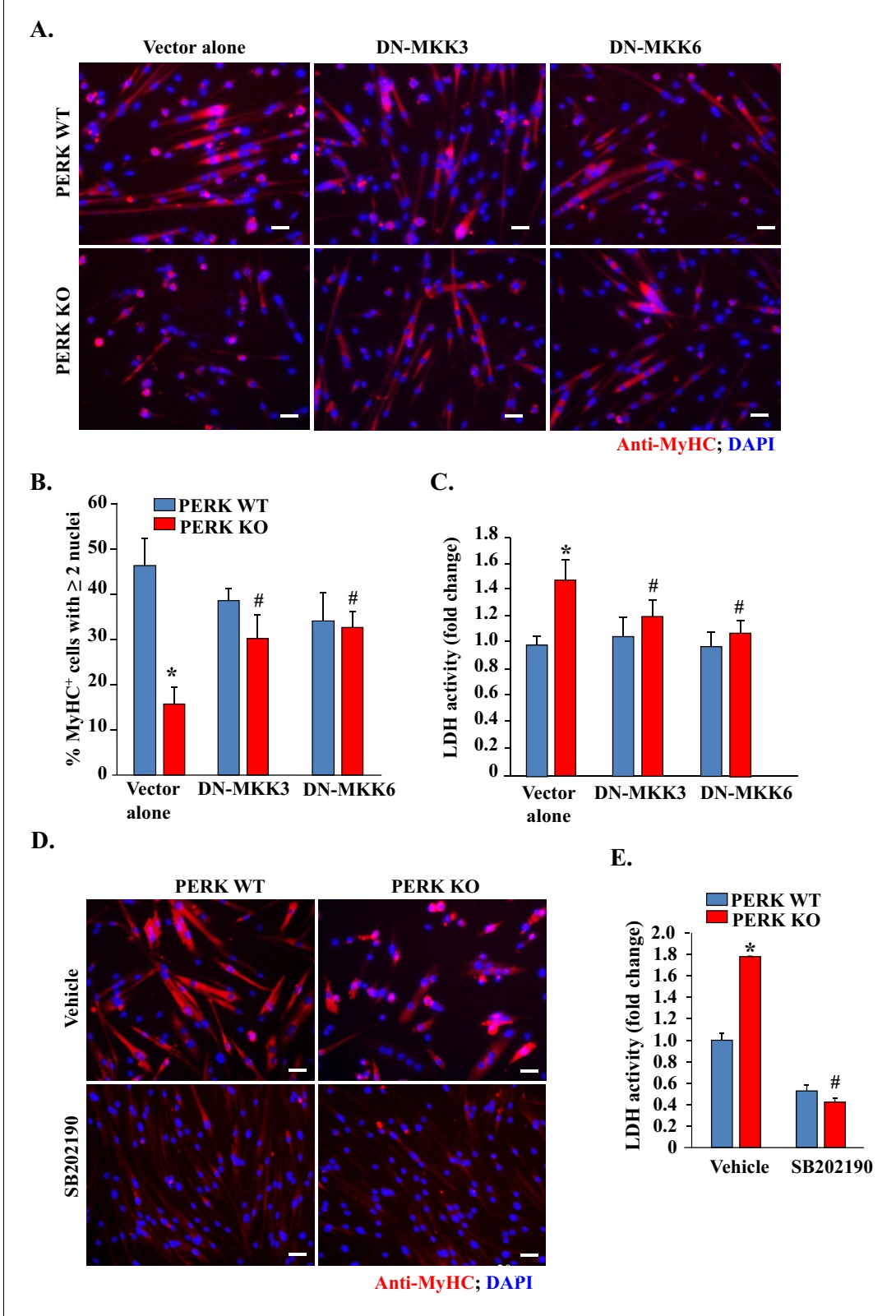

**Figure 8.** Inhibition of p38 MAPK improves myogenic cell survival and myotube formation in PERK-deficient cultures. (**A**) PERK WT and PERK KO primary myogenic cells were transfected (by electroporation) with vector alone or a dominant negative (DN) mutant of MKK3 or MKK6. The cells were incubated in DM for 48 hr and myotube formation was monitored by immunostaining for MyHC. Nuclei were visualized by staining with DAPI. Representative images presented here suggest that overexpression of DN-MKK3 or DN-MKK6 improves myotube formation in PERK KO cultures. Scale

*Figure 8 continued on next page*

*Figure 8 continued*

bars: 20 μm. (B) Quantification of percentage of MyHC$^+$ myotubes containing two or more nuclei in PERK WT and PERK KO cultures transfected with vector alone, DN-MKK3 or DN-MKK6 cDNA. (C) Relative amounts of LDH in supernatants of vector alone, DN-MKK3 or DN-MKK6 cDNA transfected PERK WT and PERK KO cultures incubated in DM for 48 hr. (D) PERK WT and PERK KO primary myogenic cells were incubated in DM for 48 hr with or without 20 μM SB202190. Representative images presented here suggest that treatment with SB202190 improved survival of MyHC$^+$ cells in PERK KO cultures. Scale bars: 20 μm. (E) Relative amounts of LDH in supernatants of PERK WT and PERK KO cultures 48 hr after addition of DM and treatment with vehicle alone or SB202190. N = 4 in each group. Data are mean ± SD. *p<0.01, values significantly different from PERK WT cultures transfected with empty vector or treated with vehicle alone by unpaired t-test. $^\#$p<0.01, values significantly different from PERK KO cultures transfected with empty vector or treated with vehicle alone by unpaired t-test.

*2001*). Interestingly, we found that genetic deletion of XBP1 in satellite cells had no effect on skeletal muscle regeneration, expression of various MRFs, or Pax7 levels suggesting that IRE/XBP1 arm of the UPR is dispensable for satellite cell-mediated regenerative myogenesis (*Figure 2* and *Figure 3I–K*). We did not find any change in the levels of ATF6 in satellite cells of injured muscle compared to uninjured muscle of WT mice (*Figure 1A*). However, a previous study has shown that the ATF6 arm of the UPR is essential for selective apoptosis of differentiation-incompetent myogenic progenitor cells during skeletal muscle development (*Nakanishi et al., 2005*). Taken together, these findings highlight that different arms of the UPR may have distinct roles in regulation of muscle progenitor cell function during embryonic development and regeneration of adult myofibers upon injury.

It was recently reported that PERK and its downstream target eIF2α are constitutively phosphorylated in freshly isolated satellite cells and their phosphorylation is reduced upon culturing for three days (*Zismanov et al., 2016*). It has also been reported that the phosphorylation of eIF2α at Serine 51 residue is essential for the self-renewal of satellite cells and overexpression of a phosphorylation resistant mutant of eIF2α (i.e. eIF2αS51A) leads to the activation and progression of satellite cells into the myogenic lineage. Satellite cells expressing the eIF2αS51A mutant are capable of undergoing differentiation and fusion with injured myofibers. While the overexpression of the eIF2αS51A mutant dramatically reduced the number of Pax7$^+$ cells in naïve muscle, only a very small reduction was noticeable by the deletion of PERK (*Zismanov et al., 2016*). However, the role of PERK in satellite cell regenerative function was not investigated. It is noteworthy that in addition to PERK, there are several other known kinases, such as double stranded RNA-activated protein kinase R (PKR), heme-regulated inhibitor eIF2α kinase (HRI), and general control nonderepressible-2 (GCN2), which can phosphorylate eIF2α in response to stress signaling (*Kilberg et al., 2009*). Furthermore, PERK also has eIF2α independent functions including the activation of an antioxidant response, calcium homeostasis, mitochondrial biogenesis, and autophagy, all of which promote cell survival in stress conditions (*Avivar-Valderas et al., 2011*; *Cavener et al., 2010*; *Lu et al., 2004*; *Mollereau et al., 2014*).

Our results demonstrate that the ablation of PERK in satellite cells inhibits regeneration of skeletal muscle in adult mice. The inhibition of muscle regeneration may be attributed to reduced activation or survival of activated satellite cells. Although we found that genetic or pharmacological inhibition of PERK significantly reduces satellite cells on cultured myofibers and in myofiber-free cultures (*Figure 4D–K*), there wasn't a significant difference in the number of Pax7$^+$ cells in skeletal muscle in naïve conditions (*Figure 4A and B*) or on freshly isolated EDL myofibers of Ctrl and P7: PERK KO mice (*Figure 4—figure supplement 1*). The absence of MyoD$^+$ cells on freshly isolated EDL myofibers of P7:PERK KO further suggests that the deletion of PERK does not affect the quiescence of satellite cells in vivo in naïve muscle (*Figure 4—figure supplement 1*). However, when we attempted to isolate and purify satellite cells from P7:PERK KO mice, most of them failed to survive and the remainder of the cells failed to expand. These findings suggest that PERK-mediated signaling is essential for the survival of satellite cells during their activation phase, especially when they are removed from their niche. During muscle injury, the niche of satellite cells is disrupted, which leads to their activation and proliferation. Indeed, we found that the number of TUNEL$^+$ cells was significantly increased in the regenerating myofibers of P7:PERK KO mice compared with corresponding Ctrl mice. Moreover, a significant increase in TUNEL$^+$ cells was noticeable in isolated myofibers from P7:PERK KO mice after 72 hr of culturing (*Figure 5*). After around 72 hr of culturing, myofiber-associated satellite cells either self-renew or differentiate into the myogenic lineage, further suggesting

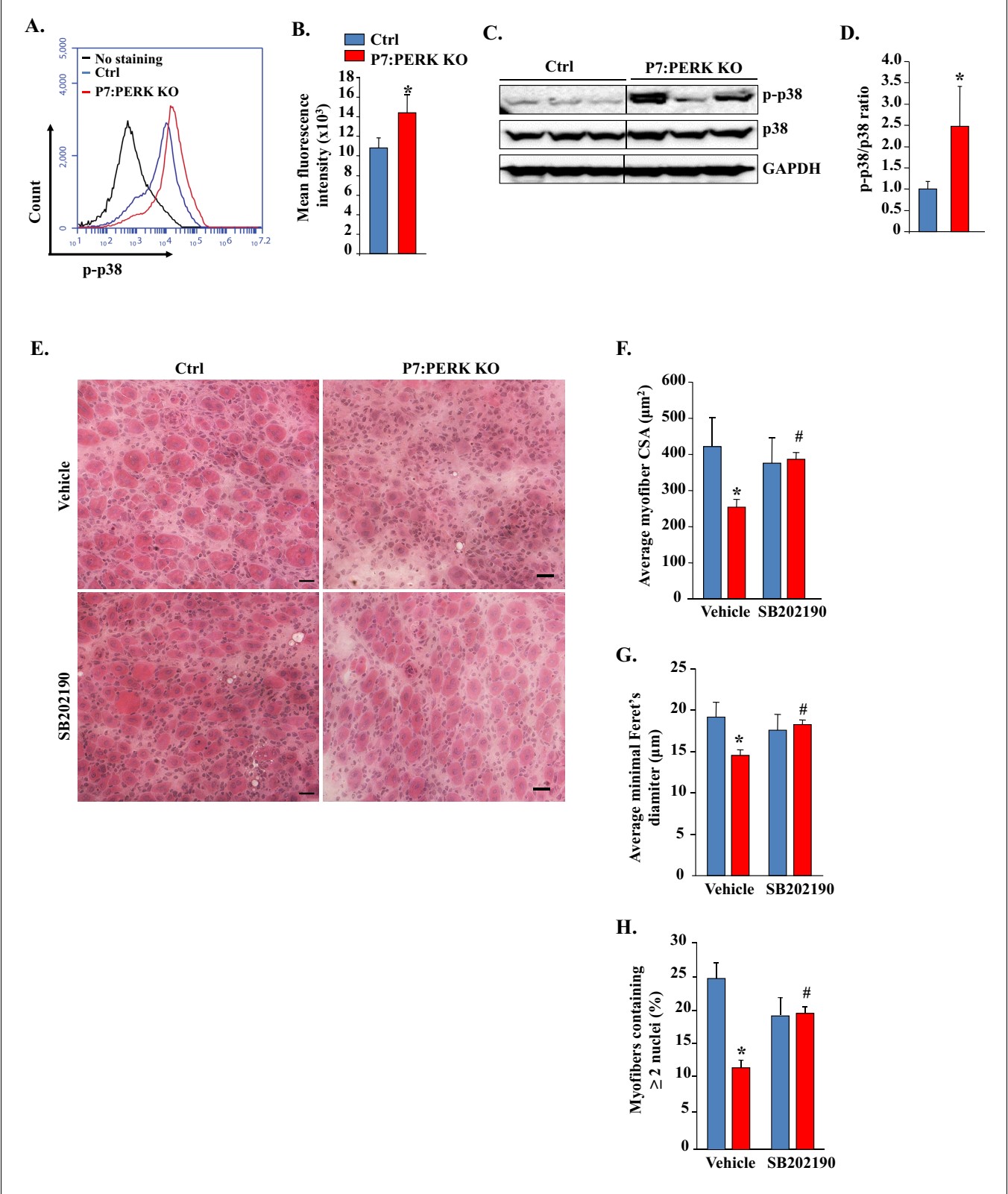

**Figure 9.** Pharmacological inhibition of p38 MAPK improves skeletal muscle regeneration in P7:PERK KO mice. (**A**) TA muscle of 3-month old Ctrl and P7:PERK KO mice was injured by intramuscular injection of 1.2% $BaCl_2$. After 5d, the muscle was collected and the single cell suspension made was analyzed for the levels of p-p38 in satellite cells. Representative histogram presented here demonstrates the levels of p-p38 in satellite cells of 5d-injured TA muscle of Ctrl and P7:PERK KO mice. (**B**) Quantification of mean florescence intensity in FACS analysis for p-p38 in satellite cells of 5d-

*Figure 9 continued on next page*

*Figure 9 continued*

injured TA muscle of Ctrl and P7:PERK KO mice. N = 4 in each group. (**C**) Representative immunoblots presented here demonstrate the levels of p-p38, total p38, and an unrelated protein GAPDH in 5d-injured TA muscle of Ctrl and P7:PERK KO mice. Black vertical line on immunoblots indicates that intervening lane has been spliced out. (**D**) Ratio of phosphorylated vs total p38 in 5d-injured TA muscle of Ctrl and P7:PERK KO mice measured by densitometric analysis of immunoblots. (**E**) TA muscle of 3-month old Ctrl and P7:PERK KO mice were injured by intramuscular injection of 1.2% BaCl$_2$ solution. The mice were also treated daily with vehicle alone or with SB202190. After 5d, the injured TA muscle was isolated and processed for H&E staining and morphometric analysis. Representative photomicrographs of H&E-stained sections illustrating that treatment with SB202190 improved myofiber regeneration in P7:PERK KO mice. Scale bar: 20 µm. Quantification of (**F**) average cross-sectional area (CSA) and (**G**) average minimal Feret's diameter of regenerating myofibers in TA muscle of Ctrl and P7:PERK KO mice. (**H**) Percentage of myofibers containing two or more centrally located nuclei per field at day 5 post injury in TA muscle of Ctrl and P7:PERK KO mice. N = 4 in each group. Data are mean ± SD. *$p<0.05$, values significantly different from corresponding Ctrl mice, as determined using unpaired Student's t-test. #$p<0.05$, values significantly different from corresponding P7: PERK KO mice treated with vehicle alone, as determined using unpaired Student's t-test.

that PERK is required for the survival of activated satellite cells which are capable of undergoing self-renewal or progression through the myogenic lineage. Similar to this study, PERK has also been found to play an important role in maintaining viability of acinar cells in exocrine pancreas. The exocrine pancreas develops and grows normally until p18 in PERK-KO mice, but then the acinar cells start to die and are completely lost within a few months (*Iida et al., 2007*).

Published reports suggest that PERK is involved in proliferation, differentiation, and survival of several cell types. PERK-deficient mice exhibit severe defects in $\beta$-cell proliferation and differentiation, resulting in low $\beta$-cell mass in pancreas at neonatal stage (*Zhang et al., 2006*). Moreover, PERK is required for skeletal development and postnatal growth (*Zhang et al., 2002*). Our results demonstrate that the inhibition of PERK reduces the differentiation of cultured myogenic cells into myotubes (*Figure 6*). Interestingly, we observed that a significantly higher proportion of PERK-deficient cells undergoes apoptosis during myogenic differentiation. These results are consistent with many published reports demonstrating that PERK plays an important role in cell survival in multiple conditions of stress (*Avivar-Valderas et al., 2011*; *Lu et al., 2004*; *Mollereau et al., 2014*; *Zhang et al., 2002*). While PERK is highly phosphorylated in satellite cells in GM, addition of DM drastically reduces p-PERK levels. Interestingly, a small increase in p-PERK and p-eIF2$\alpha$ phosphorylation and expression of the downstream target CHOP is noticeable at 6 hr after addition of DM (*Figure 6A*). This transient increase in PERK-mediated signaling appears to be essential for the initiation of the differentiation program in satellite cells. The increase in PERK activity coincides with the elevated levels of cleaved caspase-3 and cleaved PARP, the executors of apoptosis and myogenic differentiation (*Fernando et al., 2002*). It is likely that PERK-mediated signaling is a mechanism to fine-tune the survival and differentiation of myogenic cells. A previous study has shown that the incubation of myogenic cells in low serum conditions (i.e. DM) imparts significant stress on the cells and ATF6 contributes to the removal of differentiation-incompetent cells through apoptosis (*Nakanishi et al., 2005*). In contrast, our experiments suggest that the activation of PERK is a mechanism to promote the survival of myogenic cells capable of undergoing differentiation, supported by our findings that the inhibition of PERK increases the levels of cleaved capsase-3 and apoptosis in myogenic cells upon incubation in DM (*Figure 6*). Indeed, pharmacological activation of ER stress removes differentiation-incompetent myoblasts and improves the formation of myotubes (*Nakanishi et al., 2007*). Moreover, PERK-mediated up-regulation of CHOP may also regulate myogenic differentiation (*Alter and Bengal, 2011*).

The p38 MAPK is activated at later stages of myogenesis to initiate a muscle gene expression program (*Brien et al., 2013*; *Lluís et al., 2006*; *Wang et al., 2008*). However, activation of p38 MAPK in satellite cells causes precocious differentiation and diminishes their ability to undergo proliferation to repair the injured skeletal muscle (*Segalés et al., 2016*). Intriguingly, we found that the inhibition of PERK leads to the activation of p38 MAPK in myogenic cultures during differentiation without having any significant effect on the activation of other signaling pathways (*Figure 7*). Available literature suggests that p38 MAPK is one of the very important signaling pathways that gets activated in response to a variety of cellular stresses and leads to apoptosis (*Bernet et al., 2014*; *Brien et al., 2013*; *Cai et al., 2006*; *Cosgrove et al., 2014*). Our results demonstrate that the activation of p38 MAPK is an important mechanism of increased myogenic cell death and reduced

myotube formation upon inhibition of PERK both in vitro and in vivo. Inhibition of p38 MAPK improved the survival of cultured PERK KO cells (*Figure 8*) and skeletal muscle regeneration in P7: PERK KO mice (*Figure 9*). While the activation of p38 MAPK diminishes survival of PERK KO myogenic cells, it remains unknown whether p38 MAPK is directly regulated through PERK-mediated signaling or if it is a result of increased cellular stress due to an increased load of newly synthesized proteins upon inhibition of PERK. PERK promotes the antioxidant capacity of cells by directly phosphorylating the NF-E2-Related Factor 2 transcription factor, which induces the expression of various anti-oxidant genes (*Maas and Diehl, 2015*; *Motohashi and Yamamoto, 2004*). In addition, PERK has been found to regulate calcium dynamics in primary cortical neurons and pancreatic $\beta$-cells (*Wang et al., 2013*; *Zhu et al., 2016*). Indeed, a previous study has demonstrated PERK-dependent activation of JNK1/2 and p38 MAPK upon disruption of calcium homeostasis in cultured mouse embryonic fibroblasts (*Liang et al., 2006*). Therefore, it is possible that inhibition of PERK disrupts oxidative balance and calcium homeostasis which lead to myogenic cell death through the activation of p38 MAPK.

In summary, our study provides initial evidence that the PERK arm of the UPR is essential for the survival of muscle progenitor cells during myogenesis both in vivo and in vitro. Our findings provide the basis for the evaluation of PERK activity in human muscle diseases and in preclinical models of muscle degenerative disorders to determine whether increased PERK activity may prove to be beneficial.

## Materials and methods

### Animals

Satellite cell specific inducible *Eif2ak3*-knockout (i.e. P7:PERK KO) or *Xbp1*-knockout mice (i.e. P7: XBP1 KO) mice were generated by crossing *Pax7-CreER* mice (RRID:IMSR_JAX:012476) with *Eif2ak3^{fl/fl}* (RRID:IMSR_JAX:023066; *Zhang et al., 2002*) or *Xbp1^{fl/fl}* (RRID:MGI:3774017; *Hetz et al., 2008*) mice, respectively. All mice were in the C57BL6 background and their genotype was determined by PCR from tail DNA. For Cre-mediated inducible deletion of *Eif2ak3* or *Xbp1* in satellite cells, 9-week old mice were injected intraperitoneally (i.p.) with tamoxifen (10 mg per Kg body weight) in corn oil for four consecutive days. Control mice were injected with corn oil only. One week after the first injection of tamoxifen, 100 µl of 1.2% $BaCl_2$ (Sigma Chemical Co.) in saline was injected into TA muscle of mice to induce necrotic injury. For one experiment, after $BaCl_2$-mediated TA muscle injury, the mice were given daily i.p. injections of SB202190 (5 mg/kg body weight) or an equal volume of vehicle (PBS containing 50% DMSO) for five days. All experimental protocols with mice were approved in advance by the Institutional Animal Care and Use Committee (IACUC) and Institutional Biosafety Committee (IBC) of the University of Louisville.

### Histology and morphometric analysis

For the assessment of skeletal muscle morphology and regeneration, 10 µm thick transverse sections of the tibialis anterior (TA) muscle were stained with Hematoxylin and Eosin (H&E). For quantitative analysis, cross-sectional area (CSA) and minimal Feret's diameter of myofibers was analyzed in H&E-stained TA muscle sections using Nikon NIS Elements BR 3.00 software (Nikon). For each muscle, the distribution of myofiber CSA or minimal Feret's diameter was calculated by analyzing approximately 200 myofibers as described (*Paul et al., 2010*).

### Satellite cell cultures

Satellite cells were isolated from the hind limbs of 8-week-old mice as described (*Ogura et al., 2013*). PERK WT and PERK KO cells were generated by treatment of *Eif2ak3^{fl/fl}*;*Pax7-CreER* myogenic cells with vehicle alone or 0.5 µM 4-hydroxytamoxifen (Sigma Chemical Co.), respectively, for 48 hr. The 293T (Cat # CRL-3216; RRID:CVCL_0063) cell line was purchased from the American Type Culture Collection (ATCC, Manassas, Virginia) with provided information of authenticity in the year 2014. All cells were routinely tested to be free of mycoplasma contamination by DAPI staining. All the experiments with cultured satellite cells were performed in 3–4 replicates and repeated at least two times using different batches of the cells.

## Isolation and culturing of myofiber

Single myofiber cultures were established from EDL muscle after digestion with collagenase II (Worthington Biochemical Corporation, Lakewood, NJ) and trituration as described (*Dahiya et al., 2011*; *Hindi et al., 2012*). Suspended myofibers were cultured in 60 mm horse serum-coated plates in Dulbecco's modified Eagle's medium (DMEM) supplemented with 10% fetal bovine serum (FBS; Invitrogen), 2% chicken embryo extract (Accurate Chemical, Westbury, NY), 10 ng/ml basis fibroblast growth factor (Peprotech, Rocky Hill, NJ), and 1% penicillin-streptomycin for three days.

## Immunofluorescence

For the immunohistochemistry studies, frozen TA muscle sections or myofiber or myoblast cultures were fixed in 4% paraformaldehyde (PFA) in PBS, blocked in 2% bovine serum albumin in PBS for 1 hr and incubated with anti-Pax7 (1:10, DSHB Cat# pax7, RRID:AB_528428), anti-eMyHC (1:200, DSHB Cat# F1.652 RRID:AB_528358), anti-laminin (1:500, Sigma-Aldrich Cat# L9393 RRID:AB_ 477163), or anti-MyoD (1:200, Santa Cruz Biotechnology Cat# sc-304 RRID:AB_631992) in blocking solution at 4℃ overnight under humidified conditions. The sections were washed briefly with PBS before incubation with Alexa Fluor 488 (Thermo Fisher Scientific Cat# A-11034 also A11034 RRID: AB_2576217) or Alexa Fluor 594 (Thermo Fisher Scientific Cat# A-11037 also A11037 RRID:AB_ 2534095) secondary antibody for 1 hr at room temperature and then washed 3 times for 15 min with PBS. TUNEL staining was performed following a protocol from manufacturer (in situ Cell Death Detection Kit, Sigma Aldrich). Briefly, the sections or myofiber cultures were fixed in 4% paraformaldehyde and permeabilised with 0.1% Triton X-100 in 0.1% sodium citrate and incubated in TUNEL reaction mixture for 60 min at 37℃. The slides were mounted using fluorescence medium (Vector Laboratories) and visualized at room temperature on Nikon Eclipse TE 2000-U microscope (Nikon), a digital camera (Nikon *Digital Sight DS-Fi1*), and Nikon NIS Elements BR 3.00 software (Nikon). Image levels were equally adjusted using Abode Photoshop CS2 software (Adobe).

## Plasmids and gene transfer by electroporation

pCDNA3-Flag MKK6(K82A) was a gift from Roger Davis (Addgene Plasmid #13519). pRc/RSV Flag MKK3(ala) was a gift from Roger Davis (Addgene Plasmid # 14669). To overexpress specific cDNA, plasmid DNA was introduced into cells by electroporation (1500 V, 10 ms for duration, three pulses) using the Neon transfection system following a protocol suggested by the manufacturer (Invitrogen).

## Generation and use of PERK short hairpin RNA (shRNA) adenoviral vector

The target siRNA sequence for mouse *Eif2ak3* mRNA were identified using BLOCK-iT RNAi Designer online software (Life Technologies). At least 2–3 siRNA sequence were tested for efficient knockdown of target mRNA. The shRNA oligonucleotides were synthesized to contain the sense strand of target sequences for mouse *Eif2ak3* (i.e. GCAGGTCCTTGGTAATCATCA), short spacer (CTCGAG), and the reverse complement sequences followed by five thymidines as an RNA polymerase III transcriptional stop signal. Oligonucleotides were annealed and cloned into pLKO.1-Puro plasmid with AgeI/EcoRI sites. The insertion of shRNA sequence in the plasmid was confirmed by DNA sequencing. Adenovirus carrying *Eif2ak3* shRNA was generated following the manufacturer protocol (AdEasy Adenoviral Vector System, Agilent). PERK shRNA was PCR amplified from pLKO.1 plasmid and ligated into the pAdTrack-CMV vector digested at KpnI and XbaI sites. The resulted AdTrack-CMV-PERKshRNA plasmid was linearized with the PmeI and co-transformed into E. coli BJ5183 cells with the pAdEasy-1 plasmid. Clones undergoing Adtrack-Adeasy recombination were selected with kanamycin and confirmed by digestion with restriction endonuclease. The recombinant plasmid was linearized with PacI and transfected into 293 T cell line (ATCC) using Effectene Transfection Reagent (Qiagen) to package into active virus particles. Viruses were amplified by serial passage to concentrate. The titer was monitored under a microscope by visualizing the GFP marker co-expressed with PERK shRNA in the Adtrack-Adeasy recombinants.

## Western blot

Relative levels of various proteins were quantified by performing Western blot as described (*Mittal et al., 2010*; *Paul et al., 2010*). In brief, skeletal muscle of mice or cultured myoblasts or

myotubes were washed with PBS and homogenized in lysis buffer [50 mM Tris-Cl (pH 8.0), 200 mM NaCl, 50 mM NaF, 1 mM dithiothreitol, 1 mM sodium orthovanadate, 0.3% IGEPAL, and protease inhibitors]. Approximately, 50 µg protein was resolved on each lane on 10% SDS-PAGE, electrotransferred onto nitrocellulose membrane and probed using anti-MyoD (1:500, Santa Cruz Biotechnology Cat# sc-304 RRID:AB_631992), anti-myogenin (1:1000, BD Biosciences Cat# 556358 RRID:AB_396383), anti-eMyHC (1:1000, DSHB Cat# F1.652 RRID:AB_528358), anti-MyHC (1:1000, DSHB Cat# MF 20 RRID:AB_2147781), anti-phospho-PERK (1:500, Cell Signaling Technology Cat# 3179L RRID:AB_2095850), anti-PERK (1:500, Cell Signaling Technology Cat# 3192 also 3192S RRID:AB_2095847), anti-phospho-eIF2$\alpha$ (1:500, Cell Signaling Technology Cat# 3597L RRID:AB_390740), anti-eIF2$\alpha$ (1:500, Cell Signaling Technology Cat# 5324 RRID:AB_10692650), anti-CHOP (1:500, Cell Signaling Technology Cat# 5554S RRID:AB_10694399), anti-cleaved caspase-3 (1:500, Cell Signaling Technology Cat# 9664 also 9664P RRID:AB_2070042), anti-cleaved PARP (1:500, Cell Signaling Technology Cat# 5625P RRID:AB_10699460), anti-phospho-ERK1/2 (1:500, Cell Signaling Technology Cat# 9101 RRID:AB_331646), anti-ERK1/2 (1:1000, Cell Signaling Technology Cat# 9102 also 9102L, 9102S RRID:AB_330744), anti-phospho-p38 MAPK (1:500, Cell Signaling Technology Cat# 9211 RRID:AB_331641), anti-p38 MAPK (1:1000, Cell Signaling Technology Cat# 9212 RRID:AB_330713), anti-phospho-Akt (1:500, Cell Signaling Technology Cat# 4060 RRID:AB_2315049), anti-Akt (1:500, Cell Signaling Technology Cat# 9272 also 9272S RRID:AB_329827), anti-phospho-NF-$\kappa$B p65 (1:500, Cell Signaling Technology Cat# 3033 RRID:AB_331284), anti-NF-$\kappa$B p65 (1:500, Cell Signaling Technology Cat# 8242 also 8242P, 8242S RRID:AB_10859369), and anti-p100/p52 (1:500, Cell Signaling Technology Cat# 4882P RRID:AB_10828354) and detected by enhanced chemiluminescence. For loading controls, the membranes were stripped and reprobed with and anti-GAPDH (1:2000, Cell Signaling Technology Cat# 2118 also 2118L RRID:AB_561053).

## Surface sensing of translation (SUnSET) Assay

The rate of protein synthesis in myogenic cells was measured using a non-isotope labeled SUnSET method as described (*Bohnert et al., 2016*). Cultured myogenic cells were treated with vehicle alone or 1 µM GSK2606414 (Tocris Bioscience) for 12 hr followed by addition 1 µM puromycin (Sigma Chemical Co.) for 30 min. The cells were collected and protein extracts made and newly synthesized protein was detected by performing immunoblotting using primary antibody anti-puromycin (1:1000; Millipore Cat# MABE343 RRID:AB_2566826).

## Fluorescence-activated cell sorting (FACS)

Satellite cells were analyzed by performing FACS analysis as described (*Dahiya et al., 2011*; *Hindi et al., 2012*). For satellite cell isolation from heterogeneous cell population, cells were immunostained with antibodies against CD45, CD31, Sca-1, and Ter-119 for negative selection (all PE conjugated, eBiosciences), and with $\alpha$7-integrin (MBL International) for positive selection. Alexa Fluor 488 goat anti-mouse (Molecular Probes Cat# A-11029 also A11029 RRID:AB_138404) was used as a secondary antibody against $\alpha$7-integrin. To detect phosphorylated PERK or phosphorylated p38 MAPK expression in satellite cells, after labeling with antibodies against CD45 (eBioscience Cat# 12-0451-83 RRID:AB_465669), CD31 (BD Biosciences Cat# 553373 RRID:AB_394819), Sca-1 (eBioscience Cat# 12-5981-82 RRID:AB_466086), Ter-119 (eBioscience Cat# 12-5921-82 RRID:AB_466042), and $\alpha$7-integrin (MBL International Cat# K0046-3 RRID:AB_592046), the cells were fixed with 1% PFA and permeabilized using 0.2% Triton X-100. The cells were then incubated with anti-phospho-PERK (1:500, Cell Signaling Technology Cat# 3179L RRID:AB_2095850) or anti-phospho-p38 MAPK (Cell Signaling Technology Cat# 9211 RRID:AB_331641) and detected using Alexa Fluor 647 goat anti-rabbit antibody (Molecular Probes Cat# A-21245 also A21245 RRID:AB_141775). To detect CD45[+] leukocytes, cell-suspensions prepared from 5d injured muscles of Ctrl and P7:PERK KO mice were incubated with PE-conjugated CD45 antibody (eBioscience Cat# 12-0451-83 RRID:AB_465669) followed by FACS analysis. Apoptosis in cultured myogenic cells was assessed by Annexin V/propidium iodide (PI) staining followed by FACS. FACS analysis was performed on a C6 Accuri cytometer (BD Biosciences) equipped with three lasers. The output data was processed and plots were prepared using FCS Express 4 RUO software (De Novo Software).

## RNA isolation and quantitative Real-time PCR (qRT-PCR) Assay

RNA isolation and qRT-PCR were performed using a method as described (*Hindi and Kumar, 2016*; *Hindi et al., 2012*; *Ogura et al., 2015*). In brief, total RNA was extracted from skeletal muscle tissues of mice or cultured myogenic cells using TRIzol reagent (Thermo Fisher Scientific Life Sciences) and a RNeasy Mini Kit (Qiagen, Valencia, CA, USA) according to the manufacturers' protocols. First-strand cDNA for PCR analyses was made with a commercially available kit (Thermo Fisher Scientific Life Sciences). The quantification of mRNA expression was performed using the SYBR Green dye (Thermo Fisher Scientific Life Sciences) method on a sequence-detection system (model 7300; Thermo Fisher Scientific Life Sciences). Primers were designed with Vector NTI software (Thermo Fisher Scientific Life Sciences) and are available from the authors on request. Data normalization was accomplished with the endogenous control (*β*-actin), and the normalized values were subjected to a $2^{-\Delta\Delta Ct}$ formula to calculate the fold change between control and experimental groups.

## Statistical analyses

Results are expressed as mean ± standard deviation (SD). Statistical analyses used Student's *t*-test to compare quantitative data populations with normal distribution and equal variance. A value of $p < 0.05$ was considered statistically significant unless otherwise specified.

## Acknowledgements

We thank Dr. Laurie Glimcher of Cornell University for providing floxed XBP1 mice. This work was supported by funding from National Institute of Health grants AR059810, AR068313, and AG029623 to AK.

## Additional information

### Funding

| Funder | Grant reference number | Author |
|---|---|---|
| National Institute of Arthritis and Musculoskeletal and Skin Diseases | AR059810 | Ashok Kumar |
| National Institute on Aging | AG029623 | Ashok Kumar |
| National Institute of Arthritis and Musculoskeletal and Skin Diseases | AR068313 | Ashok Kumar |

The funders had no role in study design, data collection and interpretation, or the decision to submit the work for publication.

### Author contributions

GX, Conceptualization, Data curation, Formal analysis, Methodology, Writing—original draft; SMH, Conceptualization, Data curation, Formal analysis, Investigation, Writing—review and editing; AKM, YSG, Data curation, Investigation; KRB, Data curation, Investigation, Methodology; DRC, SRW, Resources, Writing—review and editing; AK, Conceptualization, Data curation, Formal analysis, Supervision, Funding acquisition, Writing—original draft, Project administration, Writing—review and editing

### Author ORCIDs

Yann S Gallot, http://orcid.org/0000-0003-4447-1448
Ashok Kumar, http://orcid.org/0000-0001-8571-2848

### Ethics

Animal experimentation: This study was performed in strict accordance with the recommendations in the Guide for the Care and Use of Laboratory Animals of the National Institutes of Health. All of the animals were handled according to approved institutional animal care and use committee

(IACUC) protocols (#13097) of the University of Louisville. All surgery was performed under anesthesia, and every effort was made to minimize suffering

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
