## [Decision Letter]

Thank you for submitting your article "The PERK arm of the UPR regulates satellite cell-mediated skeletal muscle regeneration" for consideration by *eLife*. Your article has been favorably evaluated by Sean Morrison (Senior Editor) and three reviewers, one of whom, Amy J Wagers (Reviewer #1), is a member of our Board of Reviewing Editors.

The reviewers have discussed the reviews with one another and the Reviewing Editor has drafted this decision to help you prepare a revised submission.

Summary:

This manuscript investigates the role of PERK and XBP1, components of the endoplasmic reticulum unfolded protein response, in satellite cell fate and function in muscle regeneration. The authors use a *Pax7-CreER* dependent ablation system to inducibly delete these genes in satellite cells and report that deletion of PERK causes deficits in muscle repair, associated with normal numbers of satellite cells at steady state but increased presence of apoptotic cells after muscle injury in vivo and during ex vivo culture. Interestingly, satellite cells isolated from PERK-deleted mice can grow in culture when associated with myofibers, but fail to survive in purified cultures. In addition, PERK suppression seems to impair myogenic differentiation potentially due to over-activation of p38 MAPK. The manuscript is well written and provides new information about the molecular mechanisms regulating muscle repair and muscle stem cells. However, there are a number of aspects of the paper that require further clarification. About half of these will require new experimentation, and half should be addressable through revision/clarification of the text.

Essential revisions:

1) The authors must clarify in the Methods and Results sections how old mice were when TAM was injected. At present, it is only stated that the mice were injured at 8-10 weeks of age, but it is unclear how many days before the injury TAM was administered (this matters because satellite cells do incorporate into muscle fibers during muscle growth and even at low levels at steady state, so depending on the timing of Cre deletion, there could be partial loss of function in muscle fibers as well).

2) In Figure 1, the authors should clarify what conclusion(s) can be drawn from reduced wet muscle weight in scPERK^-/-^ muscles. Would dry weights be more revealing? Stemming from the reduced wet weight, the authors should evaluate possible blunting of inflammation in this model using CD45 or F4/80 immunostaining.

3) The results in Figure 3 appear illogical as Myf5 is expressed before MyoD, which is then followed by Myogenin, however, in the figure only Myf5 and Myogenin show differences. It would seem more logical that MyoD would also be reduced as this is in the middle of this signalling pathway that leads to differentiation. Please comment on this apparent discrepancy in the text.

4) The authors should clarify whether PERK plays a role in differentiation mainly by mediating apoptosis of differentiation incompetent cells or through regulating myogenic genes. (Figure 3 shows a decrease in Myogenin levels in vivo using PERK knockouts, however PERK^-/-^ cells show no change in the levels of Myogenin and MyoD (Figure 6 with GSK2606414)). Related to this, what is the reason for reduced fiber CSA in PERK null mutants? It would be beneficial to report data as the average minimal Feret's diameter rather than CSA of muscle fibers during regeneration, since it is difficult to cut muscle at a perfect perpendicular in this tissue state.

5) Figure 4 would benefit from double staining from whole tissue sections rather than one fiber. At the moment, it would seem that there is more satellite cell self-renewal in PERK^+/+^ mice, and yet they exhibit the same number of differentiating cells. This is odd as there are more fibers being regenerated following damage in the PERK^+/+^ mice, which would likely require more differentiating cells.

6) The authors' conclusion from their enumeration of different subsets of Pax7/Myod expressing cells in culture ("Taken together, these results suggest that the inactivation of PERK diminishes the self-renewal capacity of satellite cells.") overstates the data. Their data are consistent with the notion that loss of PERK activity reduces the proportion of cells thought to be capable of self-renewal (the Pax7^+^MyoD^-^ subset), but their experiment does not assay self-renewal itself (which would require tracking of either individual cell divisions or minimally of input cell number versus output cell number). Similarly, they do not actually measure proliferation (just the fraction of cells double positive for these markers). Please re-phrase. Finally, the changes themselves are a bit subtle, given the profound effects on regeneration seen in vivo – can the authors comment on this?

7) The percentages for the apoptosis studies seem a bit conflicting with the authors' conclusion. They conclude that the self-renewing cells are the ones undergoing apoptosis (subsection “Deletion of PERK induces apoptosis in myogenic cells during regenerative myogenesis.”), but these cells represent only 9% of the cells in these cultures (Figure 4); whereas there are 15% TUNEL^+^ cells. Please reconcile this discrepancy, and define the identity of TUNEL^+^ cells in the regenerating muscle.

8) The data and interpretation in Figure 6 need further clarification. The authors show kinetics of p-PERK in control cells (Figure 6). p-PERK is apparent in undifferentiated cells and declines during differentiation. However, the transient increase in p-PERK at 6 and 12 hr post differentiation that the authors postulate coincides with increased caspase 3 activity at 12hr is not convincing. The increase in p-PERK at 6-12hr is modest, although activation of caspase 3 is robust until 48hr. Moreover, in other panels (Figure 6 and Figure 6), caspase 3 levels in control cells are minimal at 48hr in DM. The authors conclude “the transient increase in PERK signalling appears to be important for initiation of differentiation […] inhibition of PERK 24 hr after addition of DM had no effect”. The inhibition of pPERK 24 hr after addition of DM should not have an effect as there are negligible levels of pPERK at that point. Thus, the correlation of a presumed transient increase in pPERK and caspase 3 activity is unclear. Please clarify the text to address this point.

9) Is p38 signalling elevated in vivo, and does inhibition of p38 signalling impact the regeneration defect in PERK^-/-^ mutants?

10) All of the studies in Figure 7 are performed with PERK^-^ cells; however, comparison is needed to PERK^+^ in order to support the conclusion that PERK inhibition causes aberrant activation of p38 signaling (subsection “Inhibition of p38 MAPK reduces mortality and improves myotube formation in PERK-/- cultures”). In this case, one could expect the effects to be specific to PERK^-^inhibited cells, and this should be tested directly by analyzing WT vs. PERK^-^ cells in parallel.

---

## [Author Response]

*Essential revisions:*

*1) The authors must clarify in the Methods and Results sections how old mice were when TAM was injected. At present, it is only stated that the mice were injured at 8-10 weeks of age, but it is unclear how many days before the injury TAM was administered (this matters because satellite cells do incorporate into muscle fibers during muscle growth and even at low levels at steady state, so depending on the timing of Cre deletion, there could be partial loss of function in muscle fibers as well).*

Starting at the age of 9-week, the mice were given daily intraperitoneal injection of tamoxifen (TAM) for four days and one week from the first injection of TAM, TA muscle injury was performed. We had also considered reviewers’ point about the possible incorporation of PERK^-/-^ satellite cells into the muscle fiber prior to muscle injury and regeneration; therefore, we isolated myofibers from scPERK^+/+^ and scPERK^-/-^ mice and immediately fixed them for Pax7 and MyoD expression analysis in satellite cells (Figure 4—figure supplement 1). We did not find any difference in the number of myofiber associated Pax7^+^ satellite cells between the two groups at 0h post isolation. We also did not detect any MyoD^+^ satellite cells at this time point, which excludes the possibility of precocious differentiation of PERK^-/-^ satellite cells. We have also now included a schematic to show the time of TAM injection, muscle injury, and analysis of regeneration phenotype. Please refer to new Figure 1. We would also like to mention that in some initial experiments, we used 4-month old mice and obtained the same muscle regeneration defects in scPERK^-/-^ (now referred to as P7:PERK KO) mice.

*2) In Figure 1, the authors should clarify what conclusion(s) can be drawn from reduced wet muscle weight in scPERK-/- muscles. Would dry weights be more revealing? Stemming from the reduced wet weight, the authors should evaluate possible blunting of inflammation in this model using CD45 or F4/80 immunostaining.*

This figure (now Figure 1) depicts the weight of TA muscle upon isolation from tendon to tendon. The wet weight is suggestive of potential differences in muscle regeneration. After injection of myotoxin, muscle fibers undergo necrosis which leads to reduced muscle weight. The muscle weight starts increasing when the regeneration starts occurring. If muscle regeneration is inhibited, the muscle weight is generally reduced. Although this is not a conclusive phenomenon, many studies including from our group have previously used wet muscle weight as one of the criteria to study muscle regeneration (e.g. Ogura et al., Nat Commun. 2015 Dec 9;6:10123; Hong et al. J Clin Invest. 2012 Nov;122(11):3873-87). In our model, PERK is only deleted in satellite cells which contribute minimally (if any) towards inflammation. However, on reviewer’s suggestion, we investigated whether PERK deletion in satellite cells may have altered the inflammatory immune response in regenerating muscle. We have now quantified the percentage of CD45^+^ cells in TA muscle at 5d post-BaCl2-medited injury. However, there was no significant difference in the proportion of CD45^+^ cells in injured TA muscle of Ctrl and P7:PERK KO mice (Figure 1).

*3) The results in Figure 3 appear illogical as Myf5 is expressed before MyoD, which is then followed by Myogenin, however, in the figure only Myf5 and Myogenin show differences. It would seem more logical that MyoD would also be reduced as this is in the middle of this signalling pathway that leads to differentiation. Please comment on this apparent discrepancy in the text.*

There was certainly some problems in this qRT-PCR analysis. We also noticed that there was no significant increase in MyoD mRNA levels at day 5 after injury even in control mice which is a well-established phenomenon. We revisited our qRT-PCR data and found that there were some outliers and there were primer dimers in a few samples in the MyoD expression analysis. We have now repeated this experiment with a new set of mice. Consistent with published reports, we find an increase in transcript levels of MyoD in 5d-injured TA muscle of scPERK^+/+^ mice. Additionally, there is a significant decrease in mRNA levels of MyoD in 5d-injured TA muscle of P7:PERK KO compared to Ctrl mice. Moreover, the reviewers will find that our mRNA data for MRFs are consistent with their protein levels presented in Figure 3.

*4) The authors should clarify whether PERK plays a role in differentiation mainly by mediating apoptosis of differentiation incompetent cells or through regulating myogenic genes. (Figure 3 shows a decrease in Myogenin levels in vivo using PERK knockouts, however PERK^-/-^ cells show no change in the levels of Myogenin and MyoD (Figure 6 with GSK2606414)). Related to this, what is the reason for reduced fiber CSA in PERK null mutants? It would be beneficial to report data as the average minimal Feret's diameter rather than CSA of muscle fibers during regeneration, since it is difficult to cut muscle at a perfect perpendicular in this tissue state.*

We believe that PERK is required for the survival of myogenic cells capable of differentiating because we see significant more cell death in PERK^-/-^ cultures after induction of differentiation (Figure 6). PERK appears to have no direct effect on the expression of myogenic genes evidenced by our observation that the inhibition of PERK does not reduce levels of MyoD and myogenin in cultures (Figure 6). We see a reduction in MRFs (i.e. myogenin and MyoD) in regenerating myofibers of P7:PERK KO mice because the myofiber formation is reduced due to increased mortality of muscle progenitor cells available to participate in regenerative myogenesis. The injured muscle environment is a heterogeneous mix of myogenic and non-myogenic (e.g. inflammatory cells, fibroblast, fat cells, etc.) cells. A reduction in MRF’s in scPERK^-/-^ muscle can signify two possibilities as suggested by reviewers: (a) Deletion of PERK in satellite cells is directly regulating the expression of myogenic genes and therefore inhibiting differentiation; and/or (b) ablation of PERK in satellite cells is leading to myogenic cell death and therefore reducing the number of cells available to differentiate and complete the myogenic program, which will result in an overall reduction in the number and therefore percentage of myogenic cells within the injured muscle environment. For this reason, we performed the same analysis in vitro where the experimental dishes contained an equal number of pure myogenic cells and therefore any detected differences in specific RNA or protein levels will reflect a change in expression and will not be based on the proportion of a specific cell type amongst other cells as is the case in injured muscle microenvironment in vivo. If lack of PERK in satellite cells was directly regulating the expression of MRFs then we should have seen a reduction in their levels in vitro as well, however this is not the case. We observed that the levels of MRF’s were not significantly affected while the mortality of myogenic cells was significantly increased when P7:PERK KOwere incubated in differentiation medium (Figure 6). These findings lead us to conclude that myogenic cell death is the main mechanism by which muscle regeneration is compromised in P7:PERK KOmice. Stemming from those notes, we believe that the fiber CSA in 5d regenerating P7:PERK KOmuscle is reduced because there is a suboptimum number of surviving myogenic cells that will carry out the regeneration program and therefore the number of fusing myoblast per myofiber is reduced which will lead to a reduction in the fiber CSA. We also agree with the reviewers that average minimal Feret's diameter is a better measure than average CSA for such studies. We have now also included minimal Feret’s diameter wherever applicable in this revised manuscript.

*5) Figure 4 would benefit from double staining from whole tissue sections rather than one fiber. At the moment, it would seem that there is more satellite cell self-renewal in PERK^+/+^ mice, and yet they exhibit the same number of differentiating cells. This is odd as there are more fibers being regenerated following damage in the PERK^+/+^ mice, which would likely require more differentiating cells.*

We have now performed the double immunostaining for Pax7 and MyoD proteins on muscle sections from Ctrl and P7:PERK KO mice. Consistent with our analysis on isolated myofibers, we also found that the number of Pax7^+^/MyoD^-^ cells is reduced in P7:PERK KO mice TA muscle after 5 days of injury (Figure 4—figure supplement 3), which suggests a reduction in cells capable of self-renewing as stated previously. However unlike our ex vivo cultured myofiber data, we found that the number of Pax7^+^/MyoD^+^ cells were also significantly reduced in 5d injured P7:PERK KO TA muscle which is consistent with the notion that there are less myogenic cells available to participate in the myogenic program (Figure 4—figure supplement 3). As far as the seeming discrepancy in the ex vivo vs. in vivo results, this might be attributed to the variation in the corresponding myogenic environment between the two settings. It’s notable that in the injured muscle environment, factors that are intrinsic and extrinsic to the myofiber and satellite cells direct the signaling cascades to progress from pro-proliferation to pro-differentiation as regeneration ensues, which terminates with the fusion of myoblasts with each other or with the existing injured myofibers. However, in the ex vivo myofiber cultures, the myofibers and associated satellite cells are constantly cultured under proliferation conditions in growth medium, and although satellite cells do partially recapitulate the myogenic program as far as MRFs’ expression, they never completely differentiate and fuse with each other or associated myofiber within the 72h time point of culturing. Our findings indicate that PERK^-/-^ myogenic cells undergo notable cell death under differentiation conditions (as evidenced by our in vitro results in Figure 6) which might explain why myofiber-associated satellite cells cultured ex vivoare displaying a milder phenotype when compared to differentiating myogenic cultures and during in vivo muscle regeneration.

*6) The authors' conclusion from their enumeration of different subsets of Pax7/Myod expressing cells in culture ("Taken together, these results suggest that the inactivation of PERK diminishes the self-renewal capacity of satellite cells.") overstates the data. Their data are consistent with the notion that loss of PERK activity reduces the proportion of cells thought to be capable of self-renewal (the Pax7^+^MyoD^-^ subset), but their experiment does not assay self-renewal itself (which would require tracking of either individual cell divisions or minimally of input cell number versus output cell number). Similarly, they do not actually measure proliferation (just the fraction of cells double positive for these markers). Please re-phrase. Finally, the changes themselves are a bit subtle, given the profound effects on regeneration seen* in vivo *– can the authors comment on this?*

We have now removed the sentence “Taken together, these results suggest that the inactivation of PERK diminishes the self-renewal capacity of satellite cells”. Per reviewers’ suggestion we have now concluded our paragraph stating that “These findings suggest that PERK is essential for maintaining the pool of satellite cells capable of undergoing self-renewal, proliferation, and fusion with injured myofibers” We agree that the changes observed in ex vivo experiment are not dramatic but they are significant. Similar to what was mentioned above to address the previous comment, we think that the milder phenotype ex vivo is due to the fact that the myofibers are constantly cultured under proliferation conditions. Our new analysis on injured muscle sections showed that the lack of PERK drastically reduces the number of remaining and possibly surviving Pax7^+^/MyoD^-^ and Pax7^+^/MyoD^+^ cells (Figure 4—figure supplement 3), which is responsible for reduced muscle formation both in vivo and in vitro. We could have removed ex vivo myofiber results but we believe that it is important to highlight that PERK has a more important role in the survival of myogenic cells in differentiation conditions.

*7) The percentages for the apoptosis studies seem a bit conflicting with the authors' conclusion. They conclude that the self-renewing cells are the ones undergoing apoptosis (subsection “Deletion of PERK induces apoptosis in myogenic cells during regenerative myogenesis.”), but these cells represent only 9% of the cells in these cultures (Figure 4); whereas there are 15% TUNEL^+^ cells. Please reconcile this discrepancy, and define the identity of TUNEL^+^ cells in the regenerating muscle.*

We agree that the self-renewing cells associated with P7:PERK KO myofiber cultures are not enough to account for all the TUNEL^+^ cells in the same cultures. We attempted to define the identity of those TUNEL^+^ cells (both in vitro and in vivo) through repeatedly trying multiple protocols to obtain double staining for TUNEL and Pax7 or TUNEL and MyoD. However, we were unsuccessful in achieving such double staining. It seems since all three labels are localized in the nucleus, we could not obtain reliable staining for TUNEL combined with Pax7 and/or MyoD. However our new results demonstrate that regeneration-defective TA muscle of P7:PERK KO mice display a reduced number of both Pax7^+^/MyoD^-^ and Pax7^+^/MyoD^+^ atday 5 following injury, which clearly demonstrates a loss of satellite cells capable of self-renewing, proliferating, and possibly differentiating under such conditions (Figure 4—figure supplement 3). These results combined with our findings in Figure 4 and Figure 5 lead us to conclude that proliferating, differentiating, and self-renewing PERK^-/-^ satellite cells participating in myogenic regeneration undergo apoptotic cell death once muscle differentiation signaling has ensued. We have now removed the sentences that the “self-renewing cells are the ones undergoing apoptosis”.

*8) The data and interpretation in Figure 6 need further clarification. The authors show kinetics of p-PERK in control cells (Figure 6). p-PERK is apparent in undifferentiated cells and declines during differentiation. However, the transient increase in p-PERK at 6 and 12 hr post differentiation that the authors postulate coincides with increased caspase 3 activity at 12hr is not convincing. The increase in p-PERK at 6-12hr is modest, although activation of caspase 3 is robust until 48hr. Moreover, in other panels (Figure 6 and Figure 6), caspase 3 levels in control cells are minimal at 48hr in DM. The authors conclude “the transient increase in PERK signalling appears to be important for initiation of differentiation […] inhibition of PERK 24 hr after addition of DM had no effect”. The inhibition of pPERK 24 hr after addition of DM should not have an effect as there are negligible levels of pPERK at that point. Thus, the correlation of a presumed transient increase in pPERK and caspase 3 activity is unclear. Please clarify the text to address this point.*

The quality of the p-PERK immunoblot was certainly not good in our previous submission. Many times the intensity of the signal in western blotting depends on quality of antibody and may be on conditions of the cells (i.e. passage number etc.). We have now performed another independent experiment and measured the levels of pPERK and caspase-3 and other markers of ER stress and apoptosis. The reviewers will find that there is a clear increase in the pPERK levels from 6-24h after addition of DM. This coincides with the activation of caspase-3 which was previously found to be required for terminal differentiation of myogenic cells after addition of DM (Fernando et al., Proc Natl Acad Sci U S A. 2002 Aug 20;99(17):11025-30). In this report, authors demonstrate that myogenic differentiation and apoptotic cell death share similar signaling events that include but not limited to caspase-3 and PARP activation. In this section of the manuscript, our intention was to make a correlation between PERK phosphorylation and caspase-3 and PARP activation as mediators and markers of myogenic differentiation. Our findings indicate that phosphorylation of PERK is required to initiate the normal myogenic program which is marked by caspase-3 and PARP activation. We have now added a paragraph about this in the “Results” section of our revised manuscript.

The differences observed in the expression of caspase-3 activation at 48h in different blots of different figures are merely due to variation in blot exposure time and do not reflect a difference in expression at this time point. We also agree with the reviewers regarding the findings in Figure 6—figure supplement 1 of our original manuscript. Based on the kinetics of PERK phosphorylation during differentiation, it would indeed be logical that inhibition of PERK after 24h of addition of DM should not have an effect on myogenic differentiation. We performed this experiment as proof of concept for this point. However, we have now removed this data set in our revised manuscript.

*9) Is p38 signalling elevated* in vivo*, and does inhibition of p38 signalling impact the regeneration defect in PERK^-/-^ mutants?*

This is a really good point. We have now compared the levels of p38 MAPK in vivo. Consistent with our in vitro findings, we observe a significant increase in p-p38 levels in satellite cells and the whole injured TA muscle of P7:PERK KO mice compared with Ctrlmice. Using a pharmacological inhibitor of p38MAPK we have also now studied the effect of inhibition of p38 on muscle regeneration in vivo. Importantly, we found that the inhibition of p38 improves TA muscle regeneration P7:PERK KOmice. Results are presented in new Figure 9.

*10) All of the studies in Figure 7 are performed with PERK^-^ cells; however, comparison is needed to PERK^+^ in order to support the conclusion that PERK inhibition causes aberrant activation of p38 signaling (subsection “Inhibition of p38 MAPK reduces mortality and improves myotube formation in PERK^-/-^ cultures”). In this case, one could expect the effects to be specific to PERK^-^ inhibited cells, and this should be tested directly by analyzing WT vs. PERK^-^ cells in parallel.*

We have used appropriate controls in Figure 7. For example, in Figure 7, both PERK^+/+^ and PERK^-/-^ cells were used. We believe reviewers were referring to Figure 8. We used PERK^+/+^ and PERK^-/-^ cells for the results in Figure 8. However, for Figure 8, we did not show the results for PERK^+/+^ cells. This is because inhibition of p38 in control (PERK^+/+^) conditions resulted in inhibition of differentiation which is not surprising, as a physiological level of p38 upregulation is a critical requirement for adequate myogenic differentiation and a disruption in this normal biological event will interfere with the progression of the myogenic program. However, we do see a rescue in PERK^-/-^ cell differentiation following p38 inhibition because those cells express a supra-physiological level of p38 which results in cell death due to activation of apoptotic pathways. Per reviewers’ suggestion we have now included the data from the PERK^+/+^ cells under the same conditions.